# RETHINKING TRAFFIC REPRESENTATION: PRE-TRAINING MODEL WITH FLOWLETS FOR TRAFFIC CLASSIFICATION

## ABSTRACT

Network traffic classification with pre-training has achieved promising results, yet existing methods fail to represent cross-packet context, protocol-aware structure, and flow-level behaviors in traffic. To address these challenges, this paper rethinks traffic representation and proposes Flowlet-based pre-training for network analysis. First, we introduce Flowlet and Field Tokenization that segments traffic into semantically coherent units. Second, we design a Protocol Stack Alignment Embedding Layer that explicitly encodes multi-layer protocol semantics. Third, we develop two pre-training tasks motivated by Flowlet to enhance both intra-packet field understanding and inter-flow behavioral learning. Experimental results show that FlowletFormer significantly outperforms existing methods in classification accuracy, few-shot learning and traffic representation. Moreover, by integrating domain-specific network knowledge, FlowletFormer shows better comprehension of the principles of network transmission (e.g., stateful connections of TCP), providing a more robust and trustworthy framework for traffic analysis.

## 1 INTRODUCTION

Network traffic refers to data transmitted across networks, including the exchange of packets and other forms of device communication. It consists of both payload and metadata that provide critical insights into network behavior. Monitoring and analyzing traffic is essential for both network management and security (Papadogiannaki & Ioannidis, 2022; Tang et al., 2020), enabling network operators to effectively tailor resource allocation, ensure quality of service, and detect malicious activities (Gutterman et al., 2019; Hu et al., 2023; Mao et al., 2019).

Recently, pre-training methods (He et al., 2020; Zhao et al., 2023; Lin et al., 2022; Zhou et al., 2025) have achieved superior performance in traffic classification tasks. These approaches pretrain models on large volumes of unlabeled data to learn generalizable representations, which can then be fine-tuned on smaller labeled datasets for specific classification tasks.

However, despite achieving promising accuracy on given datasets, existing pre-training models for traffic classification still have significant limitations.

**First**, to balance the limited information in a single packet with the excessive length of entire flows, existing methods often design packet windows as model inputs to preserve more session context across packets. However, some designs reduce the window to a single packet, making it difficult to capture contextual semantics, while others adopt a fixed first-N packet window, which is overly rigid, hinders the modeling of intra-packet structures, and fails to cover diverse network behaviors, as shown in Figure 1a. These limitations reduce the model's ability to generalize across different traffic patterns.

**Second**, existing methods often mechanically apply NLP and CV techniques to traffic representation, such as encoding packets into 4-hex tokens with subword tokenization or reshaping flows into square images. However, these representations overlook the structural of traffic, including protocol field boundaries, hierarchical semantics, and sequential dependencies. As shown in Figure 1b, the similarity of the word embedding reveals the limited ability of the model to capture semantics, making it difficult for network operators to obtain reliable insights and interpretable representations.

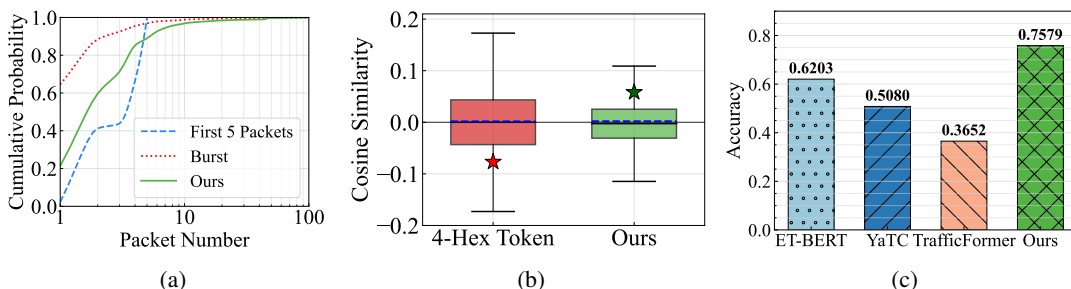

Figure 1: **Preliminary Analysis.** (a) The CDF of Packets per Packet Window. (b) Cosine Similarity of Word Embeddings. The star marks the specific similarity between ports 80 and 8080. (c) Results on Field Understanding Task (Prediction of Sequence Numbers). More details show in Appendix A

**Third**, due to the above limitations, existing pre-training tasks struggle to capture diverse traffic behavior patterns and show clear constraints in capturing semantics cross packets. We design a field understanding task that predicts key header fields of packets within a flow (here is sequence number) to evaluate whether models truly capture traffic behavior patterns. Figure 1c shows that existing methods still face considerable difficulty in understanding context within a flow, which makes their performance on downstream tasks less reliable.

To address these challenges, we propose **FlowletFormer**, a BERT-based pre-training model for network traffic analysis. Specifically, we make the following contributions:

1) We introduce **Flowlet** as a coherent behavioral unit that aggregates packets within a logical interaction. We further design **Field Tokenization** to convert each flowlet into semantically meaningful tokens based on protocol header fields.

2) We propose a **Protocol Stack Alignment-Based Embedding Layer** that explicitly encodes the hierarchical semantics of network protocols, enabling the model to distinguish fields across protocol boundaries and better capture protocol-specific behaviors.

3) We design two novel pre-training tasks motivated by our novel traffic representation. The **Masked Field Model** enhances field-level semantic understanding by predicting selectively masked critical protocol fields. The **Flowlet Prediction Task** captures logical interactions by modeling relations between Flowlets, such as HTTP requests and disconnections.

We evaluate FlowletFormer on 8 public datasets, achieving state-of-the-art performance on 7 of them, with over 6% F1 improvement on 4 datasets. Moreover, **field understanding tasks** and **word analogies similarity analysis** we propose demonstrate that FlowletFormer not only achieves higher accuracy but also better captures protocol semantics and traffic behavior than existing methods. Our code is available at `https://anonymous.4open.science/r/FlowletFormer-CC81`.

## 2 RELATED WORK

### 2.1 TRAFFIC CLASSIFICATION

Traffic classification has evolved rapidly over the past decade as networks have grown more complex and management demands have increased. Early approaches relied on packet- and flow-level statistics or rule matching, such as packet size and inter-arrival times, but these methods (Roesch, 1999; Zuev & Moore, 2005) became ineffective in encrypted environments where observable patterns are concealed. Classical machine learning methods (Taylor et al., 2016; Al-Naami et al., 2016; Panchenko et al., 2016; Sommer & Paxson, 2010) introduced classifiers such as decision trees, random forests, and SVMs, leveraging statistical summaries of flow metrics and protocol-specific characteristics. While more effective than rules, they depended heavily on feature engineering and expert knowledge. Deep learning later enabled the direct learning of high-dimensional representations from raw data. Lotfollahi et al. (2020) proposed a DNN that bypasses manual feature extraction, and subsequent work applied CNNs, RNNs, and GNNs to traffic classification (Sirinam et al., 2018; Liu et al., 2019; Shen et al., 2021; Schuster et al., 2017; Zhang et al., 2020). These models achieved strong accuracy but typically required large labeled datasets, which are costly and difficult to obtain

in practice. Moreover, traffic classification in ML and DL relies heavily on high-quality labeled datasets. Traffic data is inherently sensitive, and public datasets often contain various quality issues, such as noisy or unreliable labels (Liu et al., 2022; Engelen et al., 2021). Training on such datasets may cause models to pick up underspecification problems, including shortcut learning, overfitting to training artifacts, or learning spurious correlations, which harms their generalization (Jacobs et al., 2022; Arp et al., 2022).

## 2.2 Pre-training Methods

Due to its strong sequence modeling capability, the Transformer architecture (Vaswani, 2017) has been widely applied to network traffic classification. PERT (He et al., 2020), ET-BERT (Lin et al., 2022), TrafficFormer (Zhou et al., 2025), and PTU adopt the BERT architecture (Devlin et al., 2019) for traffic analysis, while FlowMAE (Hang et al., 2023) and YaTC (Zhao et al., 2023) employ masked autoencoders (He et al., 2022). Researchers have also explored other Transformer variants, such as T5 (Raffel et al., 2020; Wang et al., 2024a; Zhao et al., 2025) and graph-based Transformers (Van Langendonck et al., 2024). Beyond Transformers, Wang et al. (2024b) introduce the Mamba architecture for more efficient traffic analysis. Zhao et al. (2025) also revealed shortcut learnings and pitfalls of current pretraining method, including implicit flow IDs, encrypted payload, and an unfrozen encoder.

In addition to model architectures, traffic representation is a crucial component of pre-training pipelines. Raw traffic must first be transformed into a fixed format before being fed into a model. Existing approaches typically segment flows into flow segment (e.g., packets, first-N packets, or bursts), serialize these units into 4-hex strings with subword tokenization, or reshape them into structured two-dimensional matrices for training. However, these representations often misalign with the inherent characteristics of network traffic, making it difficult for pre-training methods to capture semantics, protocol structures, and sequential dependencies. This highlights the need for a new traffic representation and a corresponding pre-training model that better align with the nature of network traffic.

## 3 FlowletFormer

FlowletFormer introduces a novel framework that enables the model to capture fine-grained network behaviors and hierarchical semantics in traffic. It incorporates three key components: a new traffic representation named flowlet and field tokenization, a protocol stack alignment embedding layer to encode hierarchical structures, and two pre-training tasks tailored to flowlets.

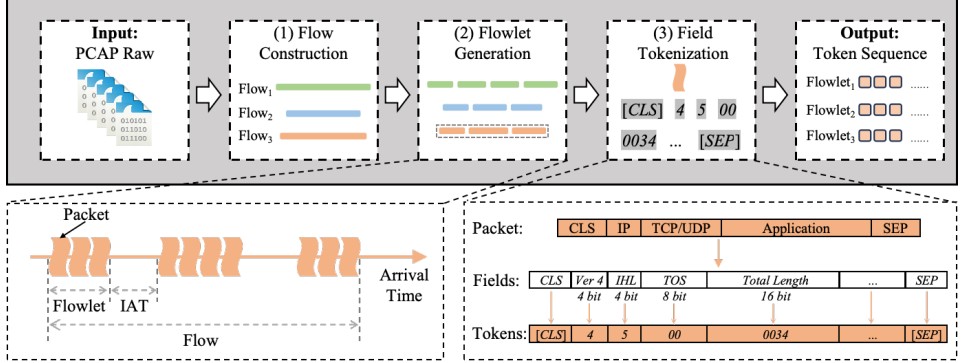

Figure 2: Flowlet and Field Tokenization.

## 3.1 Flowlet and Field Tokenization

Current pre-training models often repurpose NLP-based representations and tokenization for network traffic, overlooking its distinct structure and semantics. To address this, we propose **Flowlet and Field Tokenization**. A flowlet aggregates consecutive packets within a flow based on inter-arrival times, while field tokenization encodes each flowlet into tokens according to protocol header

boundaries. Together, they form a bridge between raw traffic and model inputs through three steps: **Flow Construction**, **Flowlet Generation**, and **Field Tokenization**, as illustrated in Figure 2.

**Flow Construction.** Raw traffic is unordered and often mixes multiple protocols, which makes pattern learning difficult. To impose semantic structure, we group packets using identical five-tuples and construct flows according to the relevant RFCs (Postel, 1981b; Eddy, 2022; Postel, 1980; 1981a). More details are provided in the Appendix B.

**Flowlet Generation.** Consider a flow $F$ consisting of a sequence of $n$ packets, denoted as $F = \{pkt_1, pkt_2, \ldots, pkt_n\}$. Each packet $pkt_i$ has an arrival timestamp $\tau_i$. The objective of Flowlet Generation is to segment this flow into multiple flowlets based on Inter-Arrival Time (IAT) between consecutive packets.

Let us define the IAT between consecutive packets as $t_i = \tau_i - \tau_{i-1}$ for $i \in 2, 3, \ldots, n$. We introduce a dynamic threshold $\theta_i$ to determine flowlet boundaries, which is adaptively adjusted based on the historical IATs. Let $W_i$ denote the IAT window up to the $i$-th packet. The threshold is calculated as:

$$\theta_i = \frac{1}{|W_i|} \sum_{t \in W_i} t \tag{1}$$

For each flowlet $\mathcal{F}_j = \{\text{pkt}_a, \text{pkt}_{a+1}, \ldots, \text{pkt}_b\}$, the inter-arrival times within the flowlet satisfy:

$$t_i \leq \theta_{i-1}, \quad \forall i \in \{a+1, \ldots, b\}. \tag{2}$$

If $\text{pkt}_b$ is the last packet of flowlet $F_j$, and $\text{pkt}_{b+1}$ is the first packet of flowlet $\mathcal{F}_{j+1}$, then:

$$t_{b+1} > \theta_b. \tag{3}$$

The algorithm begins by constructing the first flowlet from the first packet and then processes the remaining packets sequentially. When $i > 3$ and the current IAT $t_i$ exceeds the threshold $\theta_{i-1}$, a new flowlet boundary is created. Otherwise, the packet is added to the current flowlet. The algorithm continuously updates the window $W_i$ and adjusts the threshold accordingly to adapt to changing network conditions. The pseudocode is provided in the Algorithm 1.

Under this construction, flowlets serve as flow segments and coherent behavioral units, grouping packets that belong to the same logical interaction (e.g., an HTTP request–response or a media stream). By leveraging IAT to emphasize temporal correlations, flowlets ensure that packets transmitted within the same time frame are analyzed together.

**Field Tokenization.** We transform Flowlets into tokens that suitable for model input. For each packet in the flowlet, we first extract the raw bit sequences. Field tokenization then splits the sequence according to the lengths of protocol header fields, encoding the sequence into multiple hexadecimal tokens (e.g. `4` `5` `00` `0034` ...). For fields longer than two bytes and payload, we split them into multiple 4-digit hexadecimal tokens to ensure uniformity and consistency in the model input format.

In this work, we adopt word-based tokenization (Mielke et al., 2021) rather than subword methods (Chung et al., 2016; Sennrich et al., 2016; Luong & Manning, 2016), such as BPE (Sennrich et al., 2016; Gage, 1994) or WordPiece (Wu et al., 2016). The motivation is that, we treat protocol header fields as the morpheme (smallest semantic units) in traffic, similar to individual characters in Chinese. In such languages, each character is a complete and indivisible unit of meaning. Likewise, each protocol field inherently carries distinct and atomic semantics, and therefore should not be further split or processed with subword tokenization.

The maximum vocabulary size, denoted as $|V|$, is 65,812. This includes all possible tokens: 1-hex tokens (16 values), 2-hex tokens (256 values), 4-hex tokens (65,536 values), and five special tokens ([CLS], [SEP], [PAD], [MASK], [UNK]).

### 3.2 MODEL ARCHITECTURE

FlowletFormer adopts a BERT-based model architecture (Devlin et al., 2019), which consists of two modules: an Embedding Module and a Transformer Encoder Module, as illustrated in Figure 3.

**Embedding Module.** Most existing pre-training models for traffic classification directly adopt the embedding designs for NLP, including token, position, and segment embedding. However, directly

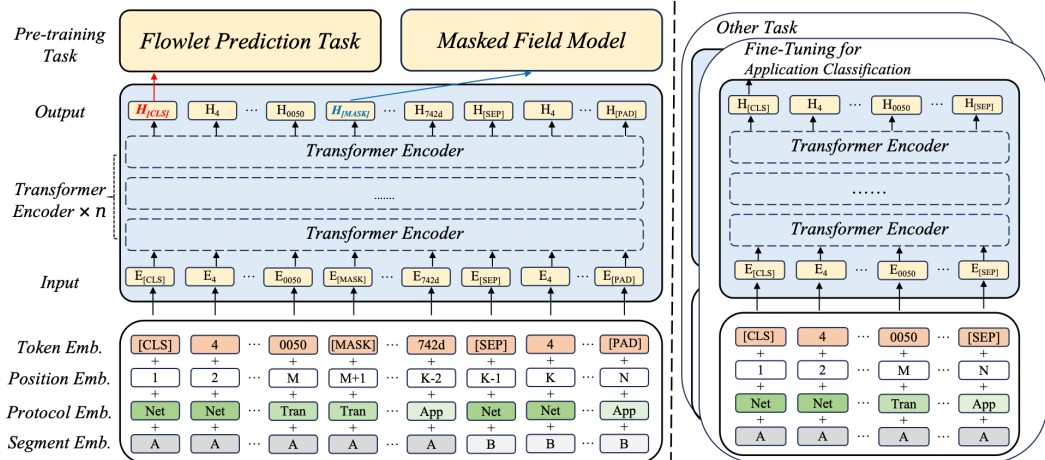

Figure 3: The flowchart of the FlowletFormer.

using these embeddings may overlook the unique characteristics of traffic. Unlike natural language, traffic exhibits a layered protocol structure with distinct forms of alignment and distribution.

Thus, we introduce a **Protocol Stack Alignment-Based Embedding Layer** into the existing embedding module. This embedding layer is specifically designed for traffic data and explicitly encodes the protocol layer associated with each token. In particular, this embedding distinguishes between the network layer, transport layer, and application layer based on the TCP/IP model (Kurose & Ross, 2001), and assigns each token an embedding corresponding to its protocol layer.

This design captures the semantic differences between different protocol layers. The model can not only process tokens based on their positions and sequential order, but also understand their functional roles within the protocol layer. This enables a hierarchical representation of traffic.

Finally, the embedding dimension is set to $D = 768$ and the input tokens are calculated by the sum of each embedding layer:

$$\mathbf{E}_{\text{input}} = \mathbf{E}_{\text{token}} + \mathbf{E}_{\text{position}} + \mathbf{E}_{\text{segment}} + \mathbf{E}_{\text{protocol}} \tag{4}$$

**Transformer Encoder Module.** FlowletFormer is built on the BERT-Based architecture and contains 12 transformer encoder layers, each with 12 multi-head self-attention heads and a position-wise feedforward network. Residual connections and layer normalization throughout the model ensure stable training and faster convergence. The total number of parameters is approximately 110 million. The number of input tokens is 512, and the dimension of each token is 768.

## 3.3 PRE-TRAINING METHOD

We introduce two novel pre-training tasks explicitly tailored to flowlet and field tokenization: the **Masked Field Model (MFM)** and the **Flowlet Prediction Task (FPT)**. These tasks are motivated by our novel traffic representation. The MFM leverages field tokenization to capture protocol-level semantics, while the FPT relies on IAT-based flowlets to model relationships between behaviorally coherent units.

**Masked Field Model.** The masked modeling task randomly masks tokens and predicts the masked. Previous studies typically use this task to learn context and dependencies. However, in network traffic, the context and dependencies carried by different tokens vary in importance. Random masking may not fully capture the structural characteristics of traffic. To address this, we design a **Masked Field Model** specifically for key fields. Instead of masking tokens uniformly at random, our approach focuses on protocol header fields that carry strong semantic and structural information.

During pre-training, 15% of the input tokens are masked. Half of these masked tokens are randomly selected from the field tokens mentioned in Table 11, while the other half are randomly selected from

the remaining tokens. For the masked tokens, we replace them with the token [MASK], a random token, or leave them unchanged with probabilities of 80 %, 10 %, and 10 %, respectively.

For masked tokens, FlowletFormer must predict the token based on the context during pre-training. The loss function used is the cross-entropy loss, as shown in Equation 5.

$$\mathcal{L}_{\mathrm{MFM}} = -\sum_{i=1}^{N} m_i \log(\hat{m_i}) \tag{5}$$

**Flowlet Prediction Task.** Flowlet is generated based on the IAT between packets, which makes Flowlet more aligned with real network interactions, providing a better representation of network behavior and traffic patterns. For example, in a file download activity, a flow may represent the entire process of downloading the file, while each Flowlet reflects specific behavior phases within the network interaction, such as the request phase, download phase, and disconnection phase.

To better capture the diverse patterns in traffic, we introduce the Flowlet Prediction Task to predict the relationships between Flowlets. During pre-training, we sample a pair of flowlets $(\mathcal{F}_A, \mathcal{F}_B)$ and form the pre-training instance. The pair is then drawn uniformly from three scenarios: $\mathcal{F}_B$ is either the immediate successor of $\mathcal{F}_A$ in the same flow (Ordered), the immediate predecessor (Swapped), or from a different flow. This design forces the model to learn intra-flow continuity, reverse-order dynamics, and clear separation of unrelated flowlets.

Unlike tasks based on individual packet or burst (Lin et al., 2022; Zhou et al., 2025), this task shifts the focus from individual packets to the relationships between behaviorally coherent Flowlets. Its goal is to capture the temporal and behavioral patterns of network traffic beyond the low-level semantics of individual packets.

Finally, the flowlet prediction task uses cross-entropy as the loss function, as shown in Equation 6.

$$\mathcal{L}_{\mathrm{FPT}} = -\sum_{i=1}^{N} y_i \log(\hat{y_i}) \tag{6}$$

Overall, the final pre-training objective is the sum of the two losses mentioned above, defined as:

$$\mathcal{L} = \mathcal{L}_{\mathrm{MFM}} + \mathcal{L}_{\mathrm{FPT}} \tag{7}$$

### 3.4 FINE-TUNING METHOD

FlowletFormer acquires generalizable knowledge during pre-training, learning diverse traffic patterns rather than being restricted to a single task. This broader understanding improves its transferability across different downstream applications.

During fine-tuning, we train the entire model architecture (Unfrozen) so that the model can effectively adapt to task-specific requirements. However, if we train only the classification head and keep the pretrained encoder Frozen, the model suffers a sharp performance drop when the downstream task contains traffic types that did not appear during pre-training. The unfrozen model is able to continue learning unseen traffic patterns while preserving its general representations.

## 4 EXPERIMENT

### 4.1 EXPERIMENT SETUP

**Pre-training Dataset.** In this work, approximately 30GB of unlabeled raw traffic data is used for pre-training. The dataset was sourced from three main repositories: ISCX-VPN2016 (NonVPN) (Draper-Gil et al., 2016), CIC-IDS2017 (Monday) (Sharafaldin et al., 2018), and the WIDE backbone dataset (January 1, 2024) (Cho et al., 2000). As shown in Table 12, these datasets encompass a significant variety of network application scenarios and protocols, such as web browsing with HTTP, file downloads with FTP, email with SMTP, and video streaming with QUIC.

During pre-training dataset construction, we consistently extract 64 consecutive bytes from the beginning of the IP layer of each packet as the model input, in order to cover key information from the IP layer and above.

**Fine-tuning Dataset.** We employ 8 datasets for fine-tuning, corresponding to 7 different downstream tasks, including **Service Type Identification** (ISCX-VPN (Service) (Draper-Gil et al., 2016) and ISCX-Tor2016 (Lashkari et al., 2017)), **Application Classification** (ISCX-VPN (App) (Draper-Gil et al., 2016)), **Website Fingerprinting** (CSTNET-TLS (Lin et al., 2022)), **Browser Classification** (Browser (Liu et al., 2019)), **Malware Classification** (USTC-TFC (Wang et al., 2017)), **Malicious Traffic Classification** (CIC-IDS2017 (Sharafaldin et al., 2018)), and **IoT Classification** (CIC-IoT2022 (Dadkhah et al., 2022)).

During fine-tuning dataset construction, we select the first five packets of each flow and extract 64 bytes starting from the IP layer of each packet. To mitigate potential biases, **we further anonymize the packets by applying IP Address&Port randomization and TCP timestamp adjustments.**

**Evaluation Metrics.** We adopt accuracy (AC), precision (PR), recall (RC), and F1 score as evaluation metrics. Further implementation details can be found in Appendix C.

### 4.2 COMPARISON WITH STATE-OF-THE-ART METHODS

We compare FlowletFormer with various baselines and state-of-the-art methods. AppScanner (Taylor et al., 2016) and CUMUL (Panchenko et al., 2016) are based on ML models. FSNet (Liu et al., 2019) and GraphDapp (Shen et al., 2021) use DL models for traffic classification. ET-BERT (Lin et al., 2022), YaTC (Zhao et al., 2023) and TrafficFormer (Zhou et al., 2025) are pre-training methods. **All pre-training methods are trained on the same pre-training and fine-tuning datasets, and the reported results are averaged over multiple runs.**

As shown in Table 1 and 2, FlowletFormer outperforms all methods on 7 datasets. Especially in the Service Type Identification (VPN, Tor) task, FlowletFormer attains an F1 score of 94% and 84%, outperforming the second-best methods(YaTC and AppScanner) by 6% and 9%, respectively. Even in the Malware Classification Task, FlowletFormer is only 0.1% lower than the best performing method (TrafficFormer) in F1 score. The results demonstrate that FlowletFormer adapts well to various traffic classification tasks and holds promise for enhancing network management and security.

Table 1: Comparison Results on ISCXVPN2016, ISCX-Tor2016, and CSTNET-TLS 1.3.

| Dataset | ISCX-VPN(Service) | | | | ISCX-Tor2016 | | | | ISCX-VPN(APP) | | | | CSTNET-TLS | | | |
|---|---|---|---|---|---|---|---|---|---|---|---|---|---|---|---|---|
| Metric | AC | PR | RC | F1 | AC | PR | RC | F1 | AC | PR | RC | F1 | AC | PR | RC | F1 |
| AppScanner | 0.8612 | 0.8678 | 0.8437 | 0.8520 | 0.8902 | 0.7715 | 0.7592 | 0.7598 | 0.7607 | 0.7036 | 0.6956 | 0.6815 | 0.7320 | 0.7129 | 0.6855 | 0.6916 |
| CUMUL | 0.6829 | 0.6747 | 0.6669 | 0.6657 | 0.7542 | 0.6471 | 0.6725 | 0.6332 | 0.5483 | 0.4442 | 0.4539 | 0.4298 | 0.5777 | 0.5336 | 0.5431 | 0.5313 |
| FSNet | 0.7679 | 0.7681 | 0.7614 | 0.7586 | 0.6705 | 0.5427 | 0.5435 | 0.5388 | 0.6576 | 0.5339 | 0.4957 | 0.4972 | 0.6537 | 0.5183 | 0.5199 | 0.4997 |
| GraphDApp | 0.6546 | 0.6270 | 0.6629 | 0.6363 | 0.7799 | 0.6168 | 0.6181 | 0.6155 | 0.4882 | 0.4143 | 0.4195 | 0.4055 | 0.6403 | 0.6017 | 0.5957 | 0.5931 |
| ET-BERT | 0.8756 | 0.8944 | 0.8525 | 0.8572 | 0.8225 | 0.7073 | 0.7375 | 0.7105 | 0.7964 | 0.7370 | 0.7013 | 0.7047 | 0.8047 | 0.7908 | 0.7777 | 0.7785 |
| YaTC | 0.9067 | 0.8991 | 0.8807 | 0.8877 | 0.8981 | 0.7384 | 0.7426 | 0.7212 | 0.8155 | 0.7599 | 0.7314 | 0.7340 | 0.8443 | 0.8404 | 0.8174 | 0.8197 |
| TrafficFormer | 0.8689 | 0.8605 | 0.8410 | 0.8373 | 0.8305 | 0.7100 | 0.6928 | 0.6932 | 0.8004 | 0.7690 | 0.7164 | 0.7221 | 0.7965 | 0.7867 | 0.7686 | 0.7675 |
| FlowletFormer | **0.9578** | **0.9539** | **0.9461** | **0.9493** | **0.9078** | **0.8411** | **0.8651** | **0.8463** | **0.8328** | **0.7859** | **0.7507** | **0.7553** | **0.8518** | **0.8506** | **0.8353** | **0.8377** |

Table 2: Comparison Results on Browser, USTC-TFC, CIC-IDS2017, and CIC-IoT2022.

| Dataset | Browser | | | | USTC-TFC | | | | CIC-IDS2017 | | | | CIC-IoT2022 | | | |
|---|---|---|---|---|---|---|---|---|---|---|---|---|---|---|---|---|
| Metric | AC | PR | RC | F1 | AC | PR | RC | F1 | AC | PR | RC | F1 | AC | PR | RC | F1 |
| AppScanner | 0.5965 | 0.5990 | 0.5926 | 0.5846 | 0.8357 | 0.8220 | 0.8478 | 0.8195 | 0.8752 | 0.9034 | 0.8964 | 0.8947 | 0.8506 | 0.8625 | 0.7780 | 0.8001 |
| CUMUL | 0.5028 | 0.5004 | 0.4990 | 0.4968 | 0.7341 | 0.5696 | 0.6518 | 0.5833 | 0.8374 | 0.7065 | 0.7337 | 0.7131 | 0.6693 | 0.6322 | 0.6479 | 0.6239 |
| FSNet | 0.5415 | 0.5559 | 0.5537 | 0.5358 | 0.8010 | 0.8177 | 0.8294 | 0.8093 | 0.8262 | 0.8405 | 0.8532 | 0.8447 | 0.8255 | 0.8158 | 0.8018 | 0.7835 |
| GraphDApp | 0.3991 | 0.4031 | 0.4067 | 0.4010 | 0.8443 | 0.8114 | 0.8198 | 0.8010 | 0.8721 | 0.8716 | 0.8527 | 0.8562 | 0.6422 | 0.5729 | 0.5900 | 0.5759 |
| ET-BERT | 0.4650 | 0.3979 | 0.4650 | 0.2680 | 0.9713 | 0.9746 | 0.9713 | 0.9715 | 0.8867 | 0.8898 | 0.8867 | 0.8830 | 0.8516 | 0.8139 | 0.8146 | 0.8088 |
| YaTC | 0.5360 | 0.5469 | 0.5371 | 0.5285 | 0.9717 | 0.9725 | 0.9716 | 0.9712 | 0.9156 | 0.9350 | 0.9156 | 0.9064 | 0.8374 | 0.8331 | 0.8095 | 0.8085 |
| TrafficFormer | 0.4750 | 0.5690 | 0.4750 | 0.2352 | **0.9758** | **0.9777** | **0.9758** | **0.9758** | 0.8894 | 0.8994 | 0.8894 | 0.8841 | 0.8678 | 0.8396 | 0.8337 | 0.8297 |
| FlowletFormer | **0.7083** | **0.7755** | **0.7083** | **0.6932** | 0.9742 | 0.9761 | 0.9742 | 0.9741 | **0.9200** | **0.9440** | **0.9200** | **0.9109** | **0.9177** | **0.8919** | **0.8820** | **0.8808** |

### 4.3 ABLATION STUDY

To evaluate the contribution of different components in FlowletFormer, we conduct an ablation study. Specifically, we systematically remove key components, including flowLet and field tokenization, the MFM, the FPT, the protocol embedding layer, and the pre-training stage. As shown in Table 3, each component contributes to the overall performance of FlowletFormer. Removing FL reduces F1 score from 0.7553 to 0.7085, and removing the MFM or the FPT lowers F1 to 0.7341 and 0.7057, respectively. These clear drops confirm their importance in capturing structural and contextual semantics. The FT and PE provides a modest yet consistent gain, suggesting its effectiveness in

modeling hierarchical semantics. Notably, removing the pre-training stage causes the most performance drop, highlighting the necessity of pre-training. More results are shown in Appendix D.

## 4.4 FEW-SHOT ANALYSIS

To further assess the effectiveness and robustness of FlowletFormer under few-shot conditions, we conduct experiments with varying data proportions. Specifically, we use the full dataset as the reference and randomly sample 40%, 20%, and 10% of the available data for few-shot training. Our few-shot evaluation on ISCX-VPN-App reveals FlowletFormer's superior data efficiency. Its maintains F1 scores of 0.8009 (40% data), 0.6224 (20%), and 0.5813 (10%). Notably, while supervised methods (e.g., CUMUL/FSNet) exhibit catastrophic performance under data scarcity, our pre-training framework maintains performance through the traffic representation model, as evidenced in Table 4. More results are shown in Appendix E.

Table 3: Ablation Study on ISCXVPN(APP). FL: Flowlet and Field Tokenization, FT: Field Tokenization, MFM: Masked Field Model, FPT: Flowlet Prediction Task, PE: Protocol Embedding Layer, and PT: Pre-Training

| Method | AC | PR | RC | F1 |
|---|---|---|---|---|
| w/o FL | 0.7872 | 0.7555 | 0.6988 | 0.7085 |
| w/o FT | 0.7994 | 0.7670 | 0.7319 | 0.7396 |
| w/o MFM | 0.8146 | 0.7604 | 0.7257 | 0.7341 |
| w/o FPT | 0.8055 | 0.7370 | 0.7021 | 0.7057 |
| w/o PE | 0.8298 | 0.7530 | 0.7348 | 0.7229 |
| w/o PT | 0.4043 | 0.2689 | 0.2678 | 0.2365 |
| FlowletFormer | **0.8328** | **0.7859** | **0.7507** | **0.7553** |

Table 4: Few-shot Analysis (F1 Score) on ISCXVPN(APP).

| Size | 100% | 40% | 20% | 10% |
|---|---|---|---|---|
| AppScanner | 0.6815 | 0.4382 | 0.5320 | 0.2222 |
| CUMUL | 0.4298 | 0.3081 | 0.2673 | 0.1550 |
| FSNet | 0.4972 | 0.4795 | 0.4752 | 0.2738 |
| GraphDApp | 0.4055 | 0.2427 | 0.2203 | 0.1944 |
| ET-BERT | 0.7047 | 0.6465 | 0.5728 | 0.4631 |
| YaTC | 0.7340 | 0.6489 | 0.5939 | 0.1805 |
| TrafficFormer | 0.7221 | 0.6085 | 0.5404 | 0.4320 |
| FlowletFormer | **0.7553** | **0.8009** | **0.6224** | **0.5813** |

## 4.5 FIELD UNDERSTANDING TASK

We introduce multiple **Field Understanding Tasks** to assess whether the model comprehends general traffic patterns. These tasks require the model to predict key header fields within a packet in a given flow. Specifically, we evaluate the comprehension of the model in four tasks: the **Flow Direction Inference** task masks the source/destination IP as well as the source/destination ports, assessing the model's ability to infer packet direction between entities based on contextual clues without direct address information; the **Transport Protocol Recognition** task focuses on masking the protocol field in the IP header, testing the model's ability to identify the transport layer protocol (e.g., TCP, UDP, ICMP); the **Sequence Awareness** task masks the sequence number and acknowledgment number within the TCP header, challenging the model to infer packet order and flow continuity; the **Connection Control Judgment** task masks the flag fields in the TCP header, which denote the state of the connection, and evaluates the model's ability to infer control signals like session establishment or termination.

These tasks evaluate the model's ability to infer direction, protocol, sequence, and control, with performance measured in three datasets: ISCX VPN, CICIDS2017, and USTC-TFC. As shown in Table 5, FlowletFormer outperforms three models in all tasks. The model's ability to effectively infer Flow Direction, Transport Protocol, Sequence Awareness, and Connection Control across diverse datasets demonstrates its strong capacity for understanding the complex behavior of network traffic.

Table 5: The Performance (Accuracy) of Pre-training Methods on Field Understanding Tasks.

| Task | Flow Direction Inference | | | Transport Protocols Recognition | | | Sequence Awareness | | | Connection Control Judgement | | |
|---|---|---|---|---|---|---|---|---|---|---|---|---|
| Dataset | VPN | IDS | TFC | VPN | IDS | TFC | VPN | IDS | TFC | VPN | IDS | TFC |
| ET-BERT | 0.4366 | 0.7096 | 0.7412 | 0.9681 | 0.9767 | 0.9981 | 0.4165 | 0.6937 | 0.6203 | 0.9041 | 0.9975 | 0.9985 |
| YaTC | 0.2617 | 0.3785 | 0.3138 | 0.1012 | 0.0858 | 0.0956 | 0.4483 | 0.6225 | 0.5080 | 0.4531 | 0.5383 | 0.3150 |
| TrafficFormer | 0.0164 | 0.1059 | 0.1128 | 0.6753 | 0.9067 | 0.8912 | 0.3659 | 0.5261 | 0.3652 | 0.3904 | 0.9983 | 0.9978 |
| FlowletFormer | **0.9313** | **0.9647** | **0.9196** | **1.0000** | **1.0000** | **1.0000** | **0.6987** | **0.7806** | **0.7579** | **0.9338** | **1.0000** | **1.0000** |

## 4.6 WORD ANALOGIES SIMILARITY ANALYSIS

In NLP, word analogy tasks assess a model's ability to capture semantic relationships between words. Through word analogy similarity analysis, we can validate whether a model has deeply understood

the semantic relationships between words. Similarly, the port number analogy analysis can be used to evaluate the pre-trained model[1], assessing its understanding of the functional and semantic relationships between network services. This capability reflects the model's deep understanding of traffic patterns acquired during pretraining, without any downstream fine-tuning.

We apply cosine similarities between the embeddings of port numbers produced by the pre-trained model to examine the relationships among common HTTP–related ports (e.g., 80, 8080, 8000). Comparing 4-hex token with our method (Table 6), we find that 4-hex token struggles to model port similarities, while FlowletFormer effectively captures these relationships, enhancing traffic classification performance. Appendix F provides more clarification.

Table 6: Port Number Analogy Cosine Similarity about Word Embedding and Input Embedding.

| Port | 80&8080 | | 80&8000 | | 8080&8000 | |
|---|---|---|---|---|---|---|
| Embedding | Word | Input | Word | Input | Word | Input |
| 4-Hex Token | -0.0768 | 0.1094 | -0.0685 | 0.1331 | **0.0740** | 0.2438 |
| Ours | **0.0582** | **0.4019** | **0.0369** | **0.3993** | 0.0400 | **0.4289** |

## 4.7 FINE-TUNING METHOD

In the fine-tuning stage, we evaluated pre-training methods and compared their performance under both Frozen and Unfrozen settings. The Frozen setup keeps the encoder parameters fixed and relies solely on the general representations learned during pre-training, serving to assess the transferability of pretrained knowledge. In contrast, the Unfrozen setup reflects the model's ability to adapt to downstream tasks, enabling it to further learn task-specific features and traffic patterns that did not appear during pre-training. This comparison provides a more comprehensive assessment of the model's generalization.

Table 7 and 8 show that FlowletFormer remains stable under the Frozen setting, with the average F1 score dropping by only 4% across four datasets. This indicates that the model has already learned transferable and general traffic patterns during pre-training. The exception is ISCX-Tor2016, where the F1 score drops by about 40% because there is no Tor traffic in pre-training dataset, leaving the model without the necessary prior knowledge when the encoder is frozen. In contrast, other pre-training baselines perform poorly in the Frozen setting, suggesting that they learn little useful generalizable representation.

Table 7: Frozen and Unfrozen Fine-tuning Results on ISCX and CSTNET.

| | ISCX-VPN(Service) | | | | ISCX-Tor2016 | | | | ISCX-VPN(APP) | | | | CSTNET-TLS | | | |
|---|---|---|---|---|---|---|---|---|---|---|---|---|---|---|---|---|
| | Frozen | | Unfrozen | | Frozen | | Unrozen | | Frozen | | Unfrozen | | Frozen | | Unfrozen | |
| | AC | F1 | AC | F1 | AC | F1 | AC | F1 | AC | F1 | AC | F1 | AC | F1 | AC | F1 |
| ET-BERT | 0.3645 | 0.2843 | 0.8756 | 0.8572 | 0.4038 | 0.2549 | 0.8225 | 0.7105 | 0.4813 | 0.2944 | 0.7964 | 0.7047 | 0.2211 | 0.1365 | 0.8047 | 0.7785 |
| YaTC | 0.3333 | 0.1667 | 0.9067 | 0.8877 | 0.1706 | 0.0498 | 0.8981 | 0.7212 | 0.2533 | 0.1328 | 0.8155 | 0.7340 | 0.0137 | 0.0047 | 0.8443 | 0.8197 |
| TrafficFormer | 0.5778 | 0.4749 | 0.8689 | 0.8373 | 0.4801 | 0.3194 | 0.8305 | 0.6932 | 0.6272 | 0.5125 | 0.8004 | 0.7221 | 0.3880 | 0.3000 | 0.7965 | 0.7675 |
| FlowletFormer | **0.8645** | **0.8466** | **0.9578** | **0.9493** | **0.5632** | **0.3806** | **0.9078** | **0.8463** | **0.7893** | **0.7181** | **0.8328** | **0.7553** | **0.6151** | **0.5614** | **0.8518** | **0.8377** |

Table 8: Frozen and Unfrozen Fine-tuning Results on Browser, USTC-TFC and CIC.

| | Browser | | | | USTC-TFC | | | | CIC-IDS2017 | | | | CIC-IoT2022 | | | |
|---|---|---|---|---|---|---|---|---|---|---|---|---|---|---|---|---|
| | Frozen | | Unfrozen | | Frozen | | Unfrozen | | Frozen | | Unfrozen | | Frozen | | Unfrozen | |
| | AC | F1 | AC | F1 | AC | F1 | AC | F1 | AC | F1 | AC | F1 | AC | F1 | AC | F1 |
| ET-BERT | 0.3450 | 0.3310 | 0.4650 | 0.2680 | 0.6700 | 0.6427 | 0.9713 | 0.9715 | 0.5628 | 0.5507 | 0.8867 | 0.8830 | 0.4589 | 0.4069 | 0.8516 | 0.8088 |
| YaTC | 0.2500 | 0.1000 | 0.5360 | 0.5285 | 0.1846 | 0.0665 | 0.9717 | 0.9712 | 0.2211 | 0.1667 | 0.9156 | 0.9064 | 0.1923 | 0.1218 | 0.8374 | 0.8085 |
| TrafficFormer | 0.4233 | 0.4035 | 0.4750 | 0.2352 | 0.8104 | 0.8090 | **0.9758** | **0.9758** | 0.6589 | 0.6551 | 0.8894 | 0.8841 | 0.5850 | 0.5381 | 0.8678 | 0.8297 |
| FlowletFormer | **0.6583** | **0.6616** | **0.7083** | **0.6932** | **0.9563** | **0.9568** | 0.9742 | 0.9741 | **0.8778** | **0.8683** | **0.9200** | **0.9109** | **0.7969** | **0.7402** | **0.9177** | **0.8808** |

## 4.8 DEEP DIVE

We further conducted in-depth evaluations of the model. In this section, we use the CSTNET-TLS dataset, as its large scale and diverse categories enable more accurate and reliable assessment.

---

[1]This is the model after pre-training but before fine-tuning, where port randomization has not been applied.

**Impact of Flowlet Threshold.** Flowlets are segmented based on IAT. Despite a mean of 1.89 seconds, the distribution is highly skewed, with most intervals much shorter. Thus, we use 0.02s, 0.2s, 2.0s, and 10s as thresholds for flowlet segmentation. Figure 4a demonstrates that threshold choice has a significant impact on downstream performance. A small threshold (e.g., 0.02s) makes nearly half of the flowlets single-packet, while a large threshold (e.g., 10s) introduces noisy long-range context. In contrast, adaptive thresholds better balance context richness and noise.

**Impact of Masked Field Ratio.** In the Masked Field Model, we select a certain proportion of specific field tokens from the mask tokens for masking, and evaluate five ratios: 10%, 30%, 50%, 70%, and 90%. Figure 4b shows that masking a moderate proportion of field tokens improves model performance, whereas excessive masking leads to performance degradation. This is because the model focuses too heavily on key fields while neglecting other information of the traffic.

**Impact of Corruption Traffic Data.** We evaluate the model under traffic corruption scenarios that may occur in real environments, considering four cases: (1) packet corruptions, (2) missing headers, (3) packet loss, and (4) header corruptions. Figure 4c shows that the model remains robust in three cases but struggles with missing headers, primarily because header loss disrupts the encoding of protocol embedding layer. More details about Deep Dive are provided in Appendix G.

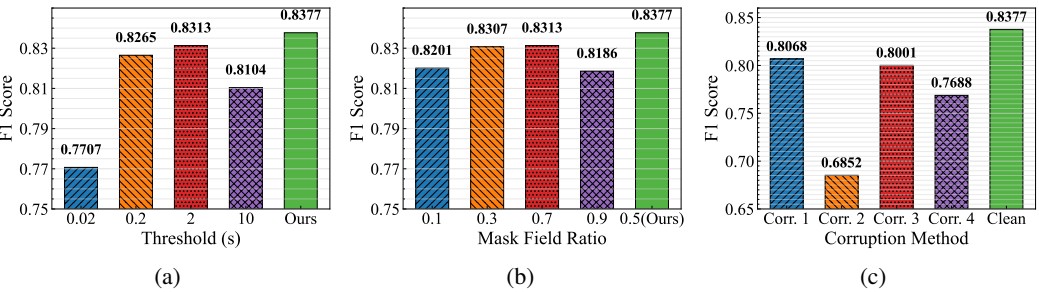

Figure 4: Deep Dive on CSTNET-TLS. (a) Sensitivity of flowlet segmentation thresholds. (b) Sensitivity of masked field ratio. (c) Results under different corruption scenarios.

## 4.9 COMPUTATIONAL COST AND COMPLEXITY

We analyze the time complexity of our method. Specifically, the complexity is: $\mathcal{O}(N \times B \times L \times (S^2 \cdot H + S \cdot H^2))$, where $N$ is the number of training steps, $B$ is the batch size, $L$ is the number of Transformer layers, $S$ is the input sequence length, and $H$ is the hidden size. We also measure the end-to-end runtimes of FlowletFormer during different phases of the train. Table 9 summarizes these results. The comparison results against other models are presented in the Appendix H.

Table 9: FlowletFormer: Computational Efficiency Across Different Phases.

| Phase | GPUs | Time | Unit/Granularity | GPU Memory (GB) |
|---|---|---|---|---|
| Pre-training | 6 | 42 h | 75.67 s / 100 steps | 28 |
| Fine-tuning | 1 | 1,153 s | 57.65 s / epoch | 17 |
| Inference | 1 | – | 150.04 samples/sec | – |

## 5 CONCLUSION

In this paper, we propose FlowletFormer, a BERT-based pre-training model designed for network traffic analysis. By introducing a Coherent Behavior-Aware Traffic Representation Model, a Protocol Stack Alignment-Based Embedding Layer, and Field-Specific and Context-Aware Pretraining Tasks, FlowletFormer effectively captures behavioral patterns, hierarchical protocol semantics, and inter-packet contextual relationships among traffic data. The experimental results demonstrate its superiority over existing methods in traffic classification.

FlowletFormer improves network traffic classification, but challenges remain. Future work includes adapting to evolving traffic patterns, enhancing robustness against adversarial attacks, incorporating multi-modal data, and optimizing computational efficiency for real-time deployment. Addressing these issues will strengthen its role in network security and traffic classification.

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

# APPENDIX

## CONTENTS OF APPENDIX

## A  Preliminary Analysis

We conduct three in-depth analyses to examine the limitations of existing method.

**First**, to balance the limited information in a single packet and the excessive length of complete flows, existing methods commonly design packet window as intermediate inputs. The intention is to retain more session context across packets while keeping the sequence length tractable for model training. However, in practice, such packet window exhibit clear limitations. We compute the cumulative distribution function (CDF) of the number of packets per window. On one hand, some packet window degenerate into single-packet units, which essentially collapses the representation back to the packet level and fails to capture any cross-packet semantics. For example, approximately 65% of bursts consist of only a single packet. This suggests that the dataset contains a high proportion of extremely short bursts, which limits the temporal context available for modeling. On the other hand, strategies that adopt a fixed number of initial packets (e.g., first-$N$ packet window) are overly rigid. These approaches cannot flexibly adapt to flows of different lengths or interaction patterns, and they ignore the variability in packet distributions across sessions.

**Second**, existing methods often adopt techniques from NLP and CV for traffic representation, such as encoding packets into 4-hex tokens with subword tokenization or reshaping traffic data into square images. However, these methods fail to align with the structure and semantics of network traffic. For example, 4-hex tokenization ignores protocol field boundaries, and network protocols' hierarchical structure is overlooked, preventing the model from capturing distinct roles of different fields. We also conducted a similarity analysis of the vocabularies in 4-hex token and our method, focusing on the word embeddings of port 80 and 8080, which both represent HTTP services. While our method correctly captures the semantic similarity between these ports, 4-hex token struggles to do so, indicating its inability to model key network relationships. This highlights a critical limitation in exist methods' semantic understanding, which FlowletFormer addresses more effectively, improving traffic classification tasks.

**Third**, as a result of the limitations discussed above, existing pretraining tasks often fail to effectively capture the diverse patterns of network traffic behavior. These methods struggle to model the semantics across packets, leading to significant constraints in their ability to learn and represent complex network interactions. To evaluate this issue, we introduce a **Field Understanding Task**, which aims to predict key header fields of packets within a flow (such as the sequence number). This task evaluates whether current models can truly capture the underlying traffic behavior patterns and understand the finer details of network communication. Field Understanding Tasks show that existing methods still face substantial challenges in capturing the context within a flow. This inability to fully grasp the flow-level semantics impacts the performance of these models on downstream tasks, making their results less reliable for network traffic analysis and prediction. Our proposed task provides a more effective way to evaluate the model's understanding of flow-level interactions, enhancing its ability to learn and generalize across various network behaviors.

## B  More Details of Our Method

### B.1  Flow Construction

To construct semantically meaningful flows from raw packet data, we apply protocol-specific rules according to standard practices outlined in RFCs and previous works. The flow construction process is based on the five-tuple: `srcIP, dstIP, srcPort, dstPort, protocol`, with additional considerations depending on the transport layer protocol.

We apply protocol-specific rules based on both packet semantics and timeout heuristics. As shown in Table 10, different protocols adopt distinct termination and reinitialization criteria. For instance, TCP flows are explicitly closed by a four-way handshake or reset flag, while UDP and ICMP rely on timeout-based or field-change-based segmentation. These rules help segment raw traffic into coherent flow units for downstream analysis.

Table 10: **Protocol-specific Rules for Flow Construction.**

| Protocol | Flow Termination Condition | New Flow Trigger |
|---|---|---|
| TCP | Four-way Handshake (FIN + FIN + ACK) Connection Reset (RST packet) Active Timeout (Flow duration exceeds 1800s) | New SYN + ACK Connection Active Timeout Expiration |
| UDP | Inactive Timeout (Flow duration exceeds 15s) | Inactive Timeout Expiration |
| ICMP | Change in ICMP `Type` Change in ICMP `Code` | Any change in `Type` or `Code` |
| Others | Flow duration exceeds 1800 seconds | Timeout Expiration |

## B.2 FLOWLET GENERATION

After flow construction, we perform the Flowlet Generation. We also describe it in Algorithm 1 The Flowlet Generation Algorithm dynamically partitions a flow into flowlets based on inter-packet arrival time. It operates as follows:

- **Initialization**: For each network flow $F = \{\text{pkt}_1, \ldots, \text{pkt}_n\}$ with timestamps $\{\tau_1, \ldots, \tau_n\}$, we compute the average inter-arrival time of the first three packets, i.e., $\theta_3 = \frac{1}{2}[(\tau_2 - \tau_1) + (\tau_3 - \tau_2)]$. This value is used as the initial threshold $\theta$ for segmentation. If $n \leq 3$, the entire flow is treated as a single Flowlet.

- **Segmentation**: For each subsequent packet $\text{pkt}_i$ $(i > 3)$, we calculate the inter-arrival time $t_i = \tau_i - \tau_{i-1}$. If $t_i > \theta_{i-1}$, we create a segmentation: the previous packet $\text{pkt}_{i-1}$ ends the current Flowlet $\mathcal{F}_j$, and $\text{pkt}_i$ begins a new one $\mathcal{F}_{j+1}$. Otherwise, $\text{pkt}_i$ is appended to the current $\mathcal{F}_j$.

- **Threshold Update**: After each decision, we update the threshold $\theta_i$ using all observed inter-arrival times up to index $i$, i.e., $\theta_i = \frac{1}{|W_i|} \sum_{t \in W_i} t$, where $W_i$ is the window of past IATs. This allows the threshold to adapt dynamically to local flow patterns.

This adaptive thresholding approach allows the segmentation process to adjust to diverse traffic dynamics. For instance, traffic patterns such as HTTP request-response cycles or video streaming often exhibit short bursts followed by longer silent gaps. By capturing such timing structures, Flowlet segmentation enables the model to better align with the logical behavior units within network communication, thus enhancing the semantic granularity of traffic representation.

---

**Algorithm 1** Flowlet Generation

---

1: **Input:** Flow $F = \{\text{pkt}_1, \ldots, \text{pkt}_n\}$ with arrival timestamps $\{\tau_1, \ldots, \tau_n\}$
2: **Output:** Flowlets $\{\mathcal{F}_1, \ldots, \mathcal{F}_k\}$
3: **Initialize:** $\mathcal{F} \leftarrow \{\text{pkt}_1\}$, $W \leftarrow \emptyset$, $\boldsymbol{flowlets} \leftarrow \emptyset$
4: **for** $i \leftarrow 2$ **to** $n$ **do**
5: $\quad t_i \leftarrow \tau_i - \tau_{i-1}$
6: $\quad$ **if** $i > 3$ **and** $t_i > \theta_{i-1}$ **then**
7: $\quad\quad$ Append $\mathcal{F}$ to $\boldsymbol{flowlets}$
8: $\quad\quad \mathcal{F} \leftarrow \{\text{pkt}_i\}$
9: $\quad$ **else**
10: $\quad\quad$ Append $\text{pkt}_i$ to $\mathcal{F}$
11: $\quad$ **end if**
12: $\quad$ Append $t_i$ to $W$
13: $\quad \theta_i \leftarrow \frac{1}{|W|} \sum_{t \in W} t$
14: **end for**
15: Append remaining $\mathcal{F}$ to $\boldsymbol{flowlets}$

---

### B.3 Key Protocol Header Fields in Masked Field Model

Table 11 lists the key fields commonly found in standard network protocols. These fields carry rich semantic and structural information that can be leveraged by traffic analysis models.

For example, fields such as port numbers and protocol types provide fundamental information about the directionality and service type of a packet, helping models distinguish between client-server roles or application types.

Sequence Number and Acknowledgment Number in the TCP header reflect the transmission order and reliability mechanisms of the protocol, offering temporal cues to infer packet sequences and session continuity.

The Total Length field, which indicates the size of an entire packet, has been demonstrated to serve as an effective signature for encrypted traffic classification in prior studies Ede-BCRDLCSP20FlowPrint, MillerHJT14.

Furthermore, TCP control flags (e.g., SYN, ACK, FIN, RST) encode connection state transitions (e.g., handshake, termination), enabling models to learn flow dynamics and session boundaries.

Similarly, ICMP's Type and Code fields identify message semantics (e.g., echo request/reply, destination unreachable), while the minimal set of fields in UDP (primarily source and destination ports) still conveys important endpoint semantics.

Table 11: Key fields in common protocol.

| Protocol | Key Fields |
|----------|-----------|
| IP | Version, Total Length, Protocol, IPID |
| TCP | Port Number, Sequence Number, Flag Acknowledgment Number, Window Size |
| UDP | Port Number |
| ICMP | Type, Code |

## C More Details in Experiment Setup

### C.1 More Details in Pre-training Dataset Construction

We describe the data preprocessing pipeline used during the pre-training stage of FlowletFormer.

**Flow Construction.** We first parsed raw PCAP files to construct flows based on five-tuples and protocol-specific rules which ensure semantically coherent flow boundaries. Each flow was saved as an individual PCAP file for subsequent processing.

**Flowlet Segmentation.** To better reflect the temporal structure and traffic behavior from application layer, we further segmented each flow into multiple flowlets. Specifically, we calculated inter-packet arrival times (IATs) and initiated a new flowlet whenever the IAT exceeded a threshold. This segmentation captures distinct behavioral units within each flow and enables the model to learn fine-grained communication patterns.

**Tokenization.** For each packet in a flowlet, we removed the Ethernet header and retained the first 64 bytes starting from the network layer. These bytes were tokenized using Field Tokenization, where individual fields in protocol headers (e.g., IP version, TTL, TCP flags) are identified and converted into semantically meaningful tokens. This tokenization approach preserves protocol semantics while producing a consistent and structured input format for the model.

Table 12 summarizes the pre-training datasets used in this work, including their sizes, number of flows, and supported protocols.

Table 12: Overview of Pre-training Datasets.

| Dataset | Size | Flow Number | Protocol |
|---|---|---|---|
| ISCX-VPN2016-NonVPN | 10.4G | 74,184 | TLS1.2, SFTP, SSDP, SNMP, NTP, MDNS, HTTP, GQUIC... |
| CIC-IDS2017-Monday | 11G | 303,436 | HTTP, HTTPS, FTP, SSH, email protocols... |
| WIDE-2024/1/1 | 9.6G | 2,322,172 | FTP, SSH, IPSec, HTTP, TLS1.2, TLS1.3, GRE, Email Protocol... |

## C.2 MORE DETAILS IN FINE-TUNING DATASET CONSTRUCTION

To ensure fair comparison and reproducibility, we describe the data preprocessing pipeline used during the fine-tuning stage of FlowletFormer.

**Data Collection and Filtering.** We collected raw PCAP files corresponding to the eight downstream tasks. Flows were constructed based on five-tuples (srcIP, dstIP, srcPort, dstPort, protocol), and each flow was saved as a separate PCAP file.

Flows were then organized by traffic category. To facilitate manageable storage and training, large files were split into smaller ones (approximately 1,000 packets each). Categories with fewer than 10 samples were discarded, and a maximum of 500 samples per class was retained to ensure balanced representation.

**Data Anonymization and Randomization.** To mitigate the risk of shortcut learning and reduce the model's dependence on protocol-specific artifacts, we performed the following anonymization steps on each flow:

- Replaced all IP addresses with randomly generated addresses;
- Randomized source and destination ports while preserving client/server roles;
- Adjusted TCP timestamps by introducing a random base time, but preserving the relative inter-packet timing.

**Tokenization.** We selected the first five packets of each flow and converted their contents to input tokens. Each packet was tokenized by retaining the first 64 tokens.

Table 13 provides an overview of all downstream tasks used for fine-tuning FlowletFormer, including dataset names, number of flows, number of classes, and example labels.

Table 13: Overview of Fine-Tuning Tasks and Datasets.

| Task | Dataset | Flow Number | Class Number | Label |
|---|---|---|---|---|
| Service Type Identification | ISCX-VPN (Service) | 1,500 | 6 | VPN-Chat,VPN-Email,VPN-Ftp... |
| | ISCX-Tor2016 | 2,922 | 8 | Audio, Browsing, Chat... |
| Application Classification | ISCX-VPN (App) | 3,289 | 10 | VPN-Youtube,VPN-Voipbuster,VPN-Vimeo... |
| Website Fingerprinting | CSTNET-TLS | 46,375 | 120 | acm.org,adobe.com,alibaba.com... |
| Browser Classification | Browser | 2,000 | 4 | Chrome,Firefox,Internet,UC |
| Malware Classification | USTC-TFC | 8,000 | 16 | Miuref,FTP,Gmail... |
| Traffic Classification | CIC-IDS2017 | 6,000 | 12 | Benign,Botnet,DDoS... |
| IoT Classification | CIC-IoT2022 | 4,931 | 12 | Attack_Flood,Idle,Interaction_Audio... |

## C.3 MORE DETAILS IN IMPLEMENTATION

In this experiment, we employ multi-GPU parallel in pre-training. A total of six GPUs are used for distributed training, with a batch size set to 16, resulting in an overall batch size of 96. The total number of training steps is 200,000, with model checkpoints saved every 10,000 steps. The Adam optimizer is chosen, with an initial learning rate of 2e-5 and a warm-up ratio of 0.1 to ensure stability during the initial stages of training.

To maintain consistency with pre-training, the fine-tuning data is processed in the same input format as the pre-training data. The packets in the flowlets are directly concatenated without [SEP] token for separation, meaning all tokens share the same segment identifiers. During the fine-tuning stage,

we select the first five packets of each network flow as the model input and extract the first 64 tokens following the Ethernet header of each packet. The dataset is split into train/validation/test sets with an 8:1:1 ratio. The model was trained for up to 20 epochs on each dataset using the AdamW optimizer with a learning rate of 6e-5, with early stopping triggered if the F1 score did not improve for 4 consecutive epochs.

The proposed method is implemented using PyTorch 2.3.1 and UER (Zhao et al., 2019) and trained on a server with 8 NVIDIA Tesla V100S GPUs.

To comprehensively evaluate the performance of classification models, we adopt widely used metrics, accuracy (AC), precision (PR), recall (RC), and F1 score (F1).

In our evaluation, precision, recall, and F1 score are macro-averaged to ensure equal consideration of all classes regardless of their frequency.

## D    MORE ABLATION STUDY

To support the figures in the main text and further illustrate the robustness of our approach, we provide complete numerical results of the ablation study across all eight downstream datasets, as shown in Table 14 and Table 15.

To thoroughly investigate the contribution of each component in **FlowletFormer**, we conducted a series of ablation experiments. The results in Table 14 and Table 15 report the performance of the full model and various degraded versions, where specific modules were removed.

**Impact of Flowlet and Field Tokenization (FL).** Removing the Flowlet and Field Tokenization module (`w/o FL`) led to significant performance drops on most datasets. In this variant, the traffic representation and tokenization revert to the burst and BPE tokenization. For example, on the ISCX-Tor2016 dataset, the accuracy decreased from 0.9078 to 0.8328 and the F1-score from 0.8463 to 0.6924. The effect is even more pronounced on the Browser dataset, where accuracy dropped from 0.7083 to 0.3700 and F1-score from 0.6932 to 0.3099. These results highlight the critical role of Flowlet segmentation and field-aware tokenization in capturing temporal dependencies and contextual coherence within sessions. By introducing Flowlets, the model learns to represent traffic in a behavior-aware manner, which facilitates more robust classification of dynamic network flows.

**Impact of Masked Field Model (MFM).** The removal of the masked field modeling task (`w/o MFM`) has dataset-specific effects. For instance, on the ISCX-VPN(Service) dataset, accuracy dropped dramatically from 0.9578 to 0.5467, indicating that MFM plays a critical role in modeling datasets with rich and structured protocol field information. It likely helps the model capture inter-field dependencies and learn which fields are important for traffic differentiation. In contrast, datasets like CSTNET-TLS and CIC-IDS2017 showed less degradation, suggesting that those tasks are less sensitive to fine-grained field semantics.

**Impact of Flowlet Prediction Task (FPT).** Removing the Flowlet Prediction Task (`w/o FPT`) caused performance degradation across several datasets, though less severe than `w/o FL` or `w/o MFM`. For example, in ISCX-Tor2016, accuracy dropped from 0.9078 to 0.8973 and F1-score from 0.8463 to 0.8052. This indicates that FPT serves as an effective auxiliary task, guiding the model to learn patterns in the temporal evolution of traffic flows, which indirectly enhances downstream classification.

**Impact of Protocol Stack Alignment-Based Embedding (PE).** The removal of the protocol embedding layer (`w/o PE`) resulted in a consistent but relatively moderate drop across datasets. This suggests that while PE enhances the model's ability to capture protocol-layer semantics, it is not the main performance bottleneck.

**Impact of Pretraining (PT).** Eliminating the pretraining stage (`w/o PT`) caused catastrophic performance degradation on all datasets. For example, on ISCX-VPN(Service), accuracy fell from 0.9578 to 0.5467 and F1-score from 0.9493 to 0.3949. These results emphasize the essential role of pretraining in learning generalizable traffic representations and initializing the model with better parameter priors for downstream tasks.

Table 14: **Ablation study results on ISCXVPN2016, ISCX-Tor2016, and CSTNET-TLS 1.3 datasets.** The abbreviations are explained as follows, FL: Flowlet and Field Tokenization, MFM: Masked Field Model, FPT: Flowlet Prediction Task, PE: Protocol Stack Alignment-Based Embedding Layer and PT: Pre-Training.

| Dataset | ISCX-VPN(Service) | | | | ISCX-Tor2016 | | | | ISCX-VPN(App) | | | | CSTNET-TLS | | | |
|---|---|---|---|---|---|---|---|---|---|---|---|---|---|---|---|---|
| Metric | AC | PR | RC | F1 | AC | PR | RC | F1 | AC | PR | RC | F1 | AC | PR | RC | F1 |
| w/o FL | 0.9133 | 0.9077 | 0.8983 | 0.8995 | 0.8328 | 0.6978 | 0.6892 | 0.6924 | 0.7872 | 0.7555 | 0.6988 | 0.7085 | 0.8025 | 0.7943 | 0.7795 | 0.7820 |
| w/o MFM | 0.5467 | 0.5429 | 0.5323 | 0.4830 | 0.4505 | 0.1790 | 0.3300 | 0.2304 | 0.8146 | 0.7604 | 0.7257 | 0.7341 | 0.8051 | 0.8024 | 0.7853 | 0.7886 |
| w/o FPT | 0.9133 | 0.8936 | 0.9138 | 0.9010 | 0.8973 | 0.8088 | 0.8145 | 0.8052 | 0.8055 | 0.7370 | 0.7021 | 0.7057 | 0.8329 | 0.8344 | 0.8162 | 0.8171 |
| w/o PE | 0.9000 | 0.9087 | 0.8656 | 0.8804 | 0.8938 | 0.8251 | 0.8145 | 0.8165 | 0.8298 | 0.7530 | 0.7348 | 0.7229 | 0.8484 | 0.8404 | 0.8323 | 0.8325 |
| w/o PT | 0.5467 | 0.4278 | 0.4278 | 0.3949 | 0.1706 | 0.0213 | 0.1250 | 0.0364 | 0.4043 | 0.2689 | 0.2678 | 0.2365 | 0.7622 | 0.7602 | 0.7357 | 0.7358 |
| FlowletFormer | **0.9578** | **0.9539** | **0.9461** | **0.9493** | **0.9078** | **0.8411** | **0.8651** | **0.8463** | **0.8328** | **0.7859** | **0.7507** | **0.7553** | **0.8518** | **0.8506** | **0.8353** | **0.8377** |

Table 15: Ablation study results on Browser, USTC-TFC, CIC-IDS2017, and CIC-IoT2022 datasets.

| Dataset | Browser | | | | USTC-TFC | | | | CIC-IDS2017 | | | | CIC-IoT2022 | | | |
|---|---|---|---|---|---|---|---|---|---|---|---|---|---|---|---|---|
| Metric | AC | PR | RC | F1 | AC | PR | RC | F1 | AC | PR | RC | F1 | AC | PR | RC | F1 |
| w/o FL | 0.3700 | 0.2787 | 0.3700 | 0.3099 | 0.9600 | 0.9680 | 0.9600 | 0.9598 | 0.8850 | 0.8870 | 0.8850 | 0.8835 | 0.8401 | 0.7881 | 0.7936 | 0.7875 |
| w/o MFM | 0.6600 | 0.6006 | 0.6600 | 0.5976 | 0.9650 | 0.9723 | 0.9650 | 0.9653 | 0.4505 | 0.1790 | 0.3300 | 0.2304 | 0.8968 | 0.8506 | 0.8543 | 0.8473 |
| w/o FPT | 0.6850 | **0.7932** | 0.6850 | 0.6428 | 0.9663 | 0.9696 | 0.9663 | 0.9658 | 0.9044 | 0.8189 | 0.9114 | 0.8429 | 0.9049 | 0.8765 | 0.8788 | 0.8736 |
| w/o PE | 0.6800 | 0.7486 | 0.6800 | 0.6745 | 0.9650 | 0.9689 | 0.9650 | 0.9648 | 0.9044 | 0.8428 | 0.9098 | 0.8653 | 0.8988 | 0.8660 | 0.8593 | 0.8587 |
| w/o PT | 0.2700 | 0.3138 | 0.2700 | 0.1387 | 0.9563 | 0.9680 | 0.9562 | 0.9571 | 0.1706 | 0.0213 | 0.1250 | 0.0364 | 0.8664 | 0.8073 | 0.8174 | 0.8089 |
| FlowletFormer | **0.7083** | 0.7755 | **0.7083** | **0.6932** | **0.9742** | **0.9761** | **0.9742** | **0.9741** | **0.9200** | **0.9440** | **0.9200** | **0.9109** | **0.9177** | **0.8919** | **0.8820** | **0.8808** |

## E  MORE FEW-SHOT ANALYSIS

To evaluate the capability of FlowletFormer under data-scarce conditions, we conduct a few-shot learning analysis. The results are reported in Table 16 and Table 17. As shown, FlowletFormer achieves competitive performance under full supervision (100% training data). More importantly, it consistently maintains relatively high F1-scores even when the amount of training data is significantly reduced.

For example, on the ISCX-VPN(Service) dataset, FlowletFormer achieves an F1-score of 0.8106 using only 10% of the training data, significantly outperforming traditional models such as App-Scanner and BIND. This indicates the strong generalization ability of FlowletFormer in few-shot settings.

However, on the Browser dataset, the performance of FlowletFormer drops more substantially under limited data, suggesting that the traffic patterns in this dataset are more complex and require more data to learn effectively.

Table 16: Few-shot Analysis (F1-score) on ISCXVPN2016, ISCX-Tor2016, and CSTNET-TLS 1.3 datasets.

| Dataset | ISCX-VPN(Service) | | | | ISCX-Tor2016 | | | | ISCX-VPN(App) | | | | CSTNET-TLS | | | |
|---|---|---|---|---|---|---|---|---|---|---|---|---|---|---|---|---|
| Size | 100% | 40% | 20% | 10% | 100% | 40% | 20% | 10% | 100% | 40% | 20% | 10% | 100% | 40% | 20% | 10% |
| AppScanner | 0.8520 | 0.7512 | 0.6074 | 0.5065 | 0.7598 | 0.7456 | 0.6195 | 0.5401 | 0.6815 | 0.4382 | 0.5320 | 0.2222 | 0.6916 | 0.6416 | 0.5661 | 0.4018 |
| CUMUL | 0.6657 | 0.5244 | 0.3873 | 0.4511 | 0.6332 | 0.5749 | 0.5252 | 0.5775 | 0.4298 | 0.3081 | 0.2673 | 0.1550 | 0.5313 | 0.4598 | 0.3659 | 0.2982 |
| FSNet | 0.7586 | 0.8384 | 0.7078 | 0.3931 | 0.5388 | 0.5426 | 0.4080 | 0.5743 | 0.4972 | 0.4795 | 0.4752 | 0.2738 | 0.4997 | 0.7132 | 0.6662 | 0.5946 |
| GraphDApp | 0.6363 | 0.5713 | 0.6137 | 0.2762 | 0.6155 | 0.5780 | 0.4622 | 0.4895 | 0.4055 | 0.2427 | 0.2203 | 0.1944 | 0.5931 | 0.4948 | 0.4372 | 0.3303 |
| ET-BERT | 0.8572 | 0.3980 | 0.2450 | 0.2583 | 0.7105 | 0.4959 | 0.3749 | 0.3512 | 0.7047 | 0.6465 | 0.5728 | 0.4631 | 0.7785 | 0.7039 | 0.6117 | 0.4819 |
| YaTC | 0.8877 | 0.0801 | 0.0721 | 0.0947 | 0.7212 | 0.6587 | 0.4994 | 0.0721 | 0.7340 | 0.6489 | 0.5939 | 0.1805 | 0.8197 | 0.7538 | 0.6375 | 0.5040 |
| TrafficFormer | 0.8373 | 0.6827 | 0.5595 | 0.3909 | 0.6932 | 0.4989 | 0.3506 | 0.3674 | 0.7221 | 0.6085 | 0.5404 | 0.4320 | 0.7675 | 0.7084 | 0.6277 | 0.5660 |
| FlowletFormer | **0.9493** | **0.8956** | **0.7356** | **0.8106** | **0.8463** | **0.7829** | **0.7166** | **0.5917** | **0.7553** | **0.8009** | **0.6224** | **0.5813** | **0.8377** | **0.8171** | **0.7273** | **0.6249** |

Table 17: Few-shot Analysis (F1-score) on Browser, USTC-TFC, CIC-IDS2017, and CIC-IoT2022 datasets.

| Dataset | Browser | | | | USTC-TFC | | | | CIC-IDS2017 | | | | CIC-IoT2022 | | | |
|---|---|---|---|---|---|---|---|---|---|---|---|---|---|---|---|---|
| Size | 100% | 40% | 20% | 10% | 100% | 40% | 20% | 10% | 100% | 40% | 20% | 10% | 100% | 40% | 20% | 10% |
| AppScanner | 0.5846 | 0.3756 | 0.3524 | 0.1838 | 0.8195 | 0.7407 | 0.6799 | 0.5733 | 0.8947 | 0.8158 | 0.7924 | 0.7265 | 0.8001 | 0.6925 | 0.5149 | 0.4027 |
| CUMUL | 0.4968 | 0.3986 | 0.3742 | 0.1500 | 0.5833 | 0.4654 | 0.3753 | 0.3631 | 0.7131 | 0.5602 | 0.5031 | 0.4991 | 0.6239 | 0.5582 | 0.5479 | 0.2113 |
| FSNet | 0.5358 | 0.4364 | 0.4444 | 0.1852 | 0.8093 | 0.6406 | 0.5563 | 0.7091 | 0.8447 | 0.7558 | 0.7244 | 0.5827 | 0.7835 | 0.5518 | 0.6089 | 0.4857 |
| GraphDApp | 0.4010 | 0.3238 | 0.2484 | 0.2875 | 0.8010 | 0.7729 | 0.6429 | 0.5219 | 0.8562 | 0.8266 | 0.6106 | 0.6531 | 0.5759 | 0.4627 | 0.3642 | 0.1766 |
| ET-BERT | 0.2680 | 0.3616 | 0.2280 | 0.2500 | 0.9715 | 0.9669 | 0.9286 | 0.8950 | 0.8830 | 0.8764 | 0.7346 | 0.7405 | 0.8088 | 0.7349 | 0.5630 | 0.4338 |
| YaTC | 0.5285 | 0.4761 | 0.4176 | 0.1613 | 0.9712 | 0.9480 | **0.9655** | 0.9159 | 0.9064 | 0.8854 | 0.6714 | 0.5902 | 0.8085 | 0.7243 | 0.7665 | 0.0758 |
| TrafficFormer | 0.2352 | 0.1520 | 0.1645 | 0.1154 | **0.9758** | **0.9703** | 0.9406 | **0.9432** | 0.8841 | 0.8725 | 0.7622 | 0.6918 | 0.8297 | 0.7578 | 0.5437 | 0.5190 |
| FlowletFormer | **0.6932** | **0.6230** | **0.6553** | **0.3095** | 0.9741 | 0.9553 | 0.9457 | 0.9380 | **0.9109** | **0.8997** | **0.8610** | **0.8510** | **0.8808** | **0.8237** | **0.8180** | **0.6152** |

## F  MORE CLARIFICATION OF WORD ANALOGIES SIMILARITY ANALYSIS

To further clarify the purpose and design of the **Word Analogies Similarity Analysis** in Section 4.6, we emphasize that this experiment is not a classification task, but rather a semantic probing analysis inspired by methodologies from natural language processing.

In NLP, analogical reasoning tasks (e.g., *"king - man + woman ≈ queen"*) are commonly used to evaluate whether pretrained language models capture meaningful token relationships. Following this intuition, we designed an analogous probing task in the context of network traffic to examine the semantic structure of token embeddings learned during pretraining.

Specifically, we selected three well-known HTTP-related port numbers (**80**, **8080**, and **8000**) and analyzed their relative positions in the learned embedding space using cosine similarity. These ports are commonly used for HTTP services and frequently co-occur in real-world traffic, thus forming a semantically coherent unit.

Our experimental results show that FlowletFormer captures the semantic similarity between these ports more accurately than baseline models. This suggests that the model has developed a deeper understanding of protocol-layer semantics and is capable of organizing related concepts (e.g., similar ports) in a meaningful embedding space.

## G  MORE DEEP DIVE

In the Deep Dive, we thoroughly analyze three key aspects: first, the impact of flowlet thresholds on downstream task performance; second, the effect of the masked field ratio on model performance; and finally, we evaluate the performance of FlowletFormer under traffic corruption scenarios.

### G.1  IMPACT OF FLOWLET THRESHOLD

To analyze the impact of the threshold, we first examine the distribution of inter-arrival times (IATs). The IATs exhibit a highly skewed distribution, with a mean of 1.89s and a large standard deviation of 36.56s. While the minimum and median values are extremely small (0 and 0.000138s, respectively), the maximum reaches nearly 1800s, indicating a heavy-tailed pattern. The quantiles further highlight this imbalance: 75% of IATs are below 0.0028s, 95% below 0.19s, and 99% below 10.22s, yet the 99.9% quantile rises sharply to 594.11s. These statistics suggest that most packet arrivals are separated by very short intervals, but a small fraction of large gaps dominate the tail, which makes threshold selection particularly sensitive.

Therefore, we select 0.02s, 0.2s, 2.0s, and 10s as thresholds for sensitivity analysis. We plotted the CDF of packets within Flowlets at different threshold values. As shown in Figure 5, when the threshold is set to 0.02s, about 60% of the Flowlets contain only a single packet, while at a threshold of 10s, only about 15% of the Flowlets contain one packet, representing two extreme cases.

We pre-trained FlowletFormer on datasets constructed with different flowlet thresholds and fine-tuned it on the same downstream task datasets. As shown in Table 18, extreme threshold values performed poorly, while moderate thresholds exhibited better performance, with our adaptive method

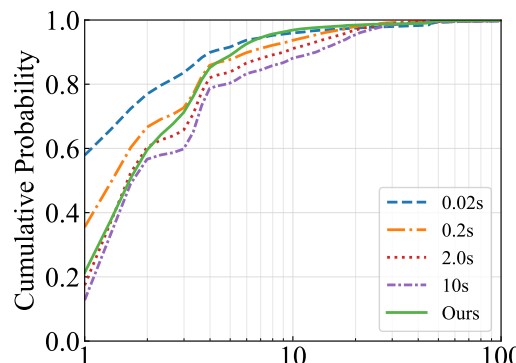

Figure 5: The CDF of Packets within Flowlets at Different Threshold

achieving the best results. This suggests that an appropriate flowlet threshold strikes a balance between capturing contextual information and suppressing noise, thereby enhancing model performance. On the other hand, extreme thresholds either lead to the loss of contextual information or introduce unnecessary noise, negatively impacting the model's learning effectiveness. Our adaptive method dynamically adjusts the threshold based on the actual data, allowing it to more accurately capture key traffic patterns and ultimately improving performance on downstream tasks.

Table 18: Performance Comparison of FlowletFormer with Different Flowlet Thresholds

| Threshold | AC | PR | RC | F1 |
|---|---|---|---|---|
| Ours | **0.8518** | **0.8506** | **0.8353** | **0.8377** |
| 0.02s | 0.8419 | 0.7902 | 0.7633 | 0.7707 |
| 0.2s | 0.8424 | 0.8445 | 0.8237 | 0.8265 |
| 2s | 0.8454 | 0.8485 | 0.8264 | 0.8313 |
| 10s | 0.8413 | 0.8346 | 0.8077 | 0.8104 |

## G.2 IMPACT OF MASKED FIELD RATIO

In the Masked Field Model, we selectively mask a proportion of specific field tokens from the total mask tokens. To assess the effect of this masking, we evaluate five different ratios: 10%, 30%, 50%, 70%, and 90%. Table 19 illustrates the relationship between the masking ratio and the performance of the model. Our findings show that masking a moderate proportion of field tokens (e.g. 30% to 50%) leads to improvements in model performance, as it allows the model to learn essential traffic patterns while still retaining a reasonable amount of contextual information. However, as the masking ratio increases, particularly beyond 70%, the performance of the model begins to degrade. This decline occurs because an excessively high proportion of key field tokens are masked, causing the model to focus too heavily on these crucial fields while ignoring other significant aspects of the traffic data. Consequently, the model loses important context and inter-field relationships, which are necessary for accurate traffic classification and understanding.

Table 19: Performance Comparison of FlowletFormer with Different Masked Field Ratio

| Threshold | AC | PR | RC | F1 |
|---|---|---|---|---|
| 0.1 | 0.8318 | 0.8492 | 0.8177 | 0.8201 |
| 0.3 | 0.8467 | 0.8425 | 0.8286 | 0.8307 |
| Ours(0.5) | **0.8518** | **0.8506** | **0.8353** | **0.8377** |
| 0.7 | 0.8506 | 0.8393 | 0.8293 | 0.8313 |
| 0.9 | 0.8366 | 0.8385 | 0.8153 | 0.8186 |

### G.3 IMPACT OF CORRUPTION TRAFFIC DATA

We evaluate the model's robustness in real-world traffic corruption scenarios that could occur in practical network environments. Specifically, we examine four types of traffic corruption: (1) packet corruption, (2) missing headers, (3) packet loss, and (4) header corruption. In type 1, we simulate a scenario where 20% of the packets in the flow experience corruption, potentially due to network interference or data transmission errors. In type 2, 20% of the packets lose their IP header, which is crucial for routing information, causing a loss of important contextual data. Type 3 simulates packet loss, where 20% of the packets are missing entirely, resulting in incomplete flow information. In type 4, 20% of the packet headers are corrupted, leading to potential misinterpretation of the protocol-specific information.

Table 20: Impact of Corruption on FlowletFormer Performance

|          | AC     | PR     | RC     | F1     |
|----------|--------|--------|--------|--------|
| Original | **0.8518** | **0.8506** | **0.8353** | **0.8377** |
| Corr. 1  | 0.8226 | 0.8218 | 0.8072 | 0.8068 |
| Corr. 2  | 0.6826 | 0.7433 | 0.669  | 0.6852 |
| Corr. 3  | 0.8154 | 0.8153 | 0.8017 | 0.8001 |
| Corr. 4  | 0.784  | 0.7897 | 0.7693 | 0.7688 |

Table 20 demonstrates that the model remains robust and performs well in the three scenarios, maintaining stable accuracy and effective traffic pattern learning. This robustness can be attributed to the model's ability to handle partial information, as it is still able to extract useful features from the remaining valid packets and headers. However, the model struggles significantly with Method 2, where headers are missing. The absence of protocol headers disrupts the encoding process in the protocol stack embedding layer, which is crucial for understanding the hierarchical structure of network traffic. This causes a sharp decline in performance, as the model loses the ability to interpret the flow's structural context properly. Our analysis highlights that while the model can handle some types of data corruption—such as packet corruption, packet loss, and partial header loss—it struggles with complete header loss, which severely impacts its ability to learn from the hierarchical structure of network protocols. This finding suggests that while the model is generally robust to real-world imperfections in traffic data, it is essential to design more resilient mechanisms for dealing with missing or corrupted headers, particularly in cases where the header plays a critical role in interpreting the traffic semantics.

## H MORE COMPUTATIONAL COST AND COMPLEXITY

Table 21 reports the full comparison of FlowletFormer against two baseline models (ET-BERT and TrafficFormer) across the three experimental phases: pretraining (6 × V100 GPUs, 200 K steps), fine-tuning (1 × V100 GPU, full epochs), and inference (throughput in samples/sec). All runs were carried out under identical hardware and configuration settings to ensure a fair evaluation of runtime, per-step/epoch granularity, and GPU memory usage.

Table 21: Computational efficiency comparison across pretraining, fine-tuning, and inference.

| Phase | Model | GPUs | Time | Unit/Granularity | GPU Memory (GB) |
|-------|-------|------|------|------------------|-----------------|
| Pretraining | FlowletFormer | 6 | 42 h | 75.67 s / 100 steps | 28 |
|  | ET-BERT | 6 | **41 h** | **73.87 s / 100 steps** | 28 |
|  | TrafficFormer | 6 | 45 h | 82.00 s / 100 steps | 28 |
| Fine-tuning | FlowletFormer | 1 | **1,153 s** | **57.65 s / epoch** | 17 |
|  | ET-BERT | 1 | 1,177 s | 58.85 s / epoch | 17 |
|  | TrafficFormer | 1 | 1,158 s | 57.90 s / epoch | 17 |
| Inference | FlowletFormer | 1 | — | 150.04 samples/sec | — |
|  | ET-BERT | 1 | — | **148.92 samples/sec** | — |
|  | TrafficFormer | 1 | — | 150.45 samples/sec | — |

# I  LIMITATION

Though FlowletFormer achieves fine-grained behavioral analysis within each flowlet, it still has several limitations.

First, the fixed maximum input length forces us to split long flows into shorter flowlets. While this enables detailed study of intra-flow behaviors, it prevents the model from learning unified patterns over entire long flows, which may be crucial for detecting certain sophisticated or slow-evolving anomalies.

Second, our Field Tokenization treats each protocol field as an independent "word" analogous to treating every single Chinese character as a separate token. Although this captures the finest-grained units, it cannot model semantic entities that span multiple fields. In future work, we could adopt Chinese word segmentation techniques to merge common adjacent fields into higher-level tokens

Third, because FlowletFormer is based on the BERT architecture, both pretraining and real-time inference demand substantial GPU resources. This high computational and memory overhead may limit deployment in resource-constrained environments or scenarios requiring very high throughput.

Lastly, despite introducing protocol-stack alignment and field-aware pretraining objectives, the internal decision process of FlowletFormer remains difficult to interpret and audit. This lack of transparency can be problematic in high-security settings where explainability and trust are paramount.

# J  THE USE OF LARGE LANGUAGE MODELS (LLMS)

During the preparation of this manuscript, we employed a large language model (LLM) to assist with language refinement. In the early stages, the LLM was used for grammar and spelling checks as well as automatic corrections. At later stages, it was consulted to polish certain sentences for improved clarity, readability, and academic style. All outputs were carefully reviewed and refined by the authors. Importantly, the LLM was not used to generate ideas, conduct experiments, perform analyses, or draw conclusions.

**Broader Impacts**  While FlowletFormer can significantly enhance the accuracy of anomaly detection and threat mitigation, thereby contributing to more secure and reliable networks, it also carries potential risks. On the positive side, better traffic classification aids in detecting malicious activities (e.g., DDoS, malware propagation) and supports privacy-preserving analytics by filtering out sensitive flows before further processing. On the negative side, the same techniques could be repurposed for intrusive traffic monitoring or profiling of users, raising privacy and ethical concerns. To mitigate such risks, we advocate for transparent deployment policies, strict access controls, and regular audits of model usage.

