# OpenReview forum: "Rethinking Traffic Representation: Pre-training Model with Flowlets for Traffic Classification"
_ICLR.cc/2026/Conference — Submitted to ICLR 2026_

### Official Review · Reviewer_LKds · 2025-10-30

**Soundness:** 1
**Presentation:** 2
**Contribution:** 1
**Rating:** 0
**Confidence:** 5

**Summary:**

One of the many papers tackling traffic analysis with a BERT-like (with very tiny engineering changes) approach. The paper follows the same line an present very minor incremental detail, with no actual methodological contribution.

Several of these paper and the dataset they use have been found to be flawed and debunked in several rank A* conferences -- starting from ACM CCS'2022 and with an up-to-date debuning in ACM SIGCOMM'2025

This ignorance can be either be done on purpose (which is bordeline  unhetical... how can they cite a paper from CCS'2018 and ignore *that* paper from CCS'2022? Same for citing SIGCOMM'2019 and ignoring *that* one in  SIGCOMM'2025?) or in full unawararennss of such work (which is equally worring and disqualify the authors work altogether), but the result does not change.


Additionally, the engineering is lightweight, the evaluation is biased and flawed: the paper would not be accepted at one of the above rank A* conference of network/security field.

Finally, there is no methodological contribution, there is no statistical relevance: as such, the paper does not find its place in ICLR'26 either



[CCS 2022] Jacobs, Arthur S., et al. "Ai/ml for network security: The emperor has no clothes." Proceedings of the 2022 ACM SIGSAC Conference on Computer and Communications Security. 2022.


[USENIX Sec 2022] Arp, Daniel, et al. "Dos and don'ts of machine learning in computer security." 31st USENIX Security Symposium (USENIX Security 22). 2022.


[SIGCOMM 2025]
Zhao, Yuqi, et al. "The Sweet Danger of Sugar: Debunking Representation Learning for Encrypted Traffic Classification." Proceedings of the ACM SIGCOMM 2025 Conference. 2025.

**Strengths:**

The paper has more flaws than strenghts -- but if I have to find one, then I would say the paper is clearly written (although I don't agree with the  styling m

**Weaknesses:**

all weaknesses are detailed next in the **Question** section

- Incremental in nature
- No learning methodological contribution
- Flowlet Lightweight engineering contribution
- Tokenizer Lightweight engineering contribution
- Possibly biased, weak evaluation, which translate into fundamentally
- Lack of statistical relevance
- lack of critical results analysis
- lack of comparison baselines
- lack of relevant technical details

**Questions:**

This paper has several flaws that prevent it from publishing



## Incremental in nature

 add itself to the pile of paper in the comparison table

## No learning methodological contribution

 BERT-based with ``automatic'' definition of flowlet based on interarrival time (IAT) and some loose considerations about tokenizer (flawed in my opinion) or loss (too narrow, specific and not a contribution per se)

## Flowlet Lightweight engineering contribution

defining a flowlet by using an IAT filter (Alg in B2 p 16) is very lightweight. it is a bonus not having to define the number of packets in a burst a priori as generally done, but it does not qualify as a scientific/technical contribution (doublly so given the evaluation flaws)

##  Tokenizer Lightweight engineering contribution

 the paper states that related work overlooks the nature of the protocol segmentation. it pompously says to use morphenes to have undividied semnantic units -- but the true fact is that their tokenizer is not doing that at all. several fields in IP, TCP and any protocols are binary flags, which are independent yet packed together in the same byte for transfer, whose smallest units would be a bit and not a hex-unit (packing 4 such flags). So adding sub 16 sub encodings to a 65k vocabulary does not ring as a fundamental contribution unless you would be able to show instances (not rand% accuracy) where this does a semantic contribution -- yet even in Appx D. FL does joint Flowlet and tokenization

## Possibly biased, weak evaluation, which translate into fundamentally flawed study.

Starting from ACM CCS'2022 and with an up-to-date debuning in ACM SIGCOMM'2025, researchers have shown limits of these approaches attempting at learning from encrypted traffic payload. These studies are widely known,
and additionally suggest (along with USENIX Sec'22)  best practices that this work do not follow. At the end, the gap with the proposal and the simple baselines from ACM CCS'2022 and ACM SIGCOMM'2025 suggest this work is trying to shoot a mosquito with a cannon



## Lack of statistical relevance

 all tables dumps 4 decimal values, no statistical relevance whatsoever -- no mean/ci, no repetittions, no statistical tests,  no paired tests, no critical distance plots.

## lack of critical results analysis

some of the results show 1.0000 (Tab5)-- there is no critical analysis of the results whatsovere, likely some shortcut as those already shown in [CCS22] for the ISCX VPN-nonVPN and CIC-IDS-2017  datasets -- a 5 nodes DecisionTree achieves in excess of 99% accuracy for those tasks due to shortcut learning, which was debunked 5 years ago already

whereas shortcut learning are  mentioned in appendix C.2, it is not enough (randomized IP addresses and ports, and removed aboslute timestaps)

## lack of comparison baselines

benchmnark for simple (eg given SEQNO sequence, have a specialized ML for
that taks, and show the gap) as done in CCS, SIGCOMM

## lack of relevant technical details

fine tuning -- is it end-to-end (=destroying pre-training value)
or layers are frozen -- check SIGCOMM'25 why this is relevant

---

> ### Author Response · Authors · 2025-11-18
>
> We appreciate the reviewer’s time, but we disagree with your claims that misrepresent both our methodology and the current state of the field. Below we clarify factual inaccuracies and respond point-by-point.
>
> ****
>
> **For "Ignore CCS’22 / USENIX’22 / SIGCOMM’25"**, CCS’22 and USENIX Sec’22 focused on shortcut learning and pitfalls in supervised ML, which are outside the scope of our original paper. And these two works did not discuss and cite **any** BERT-based traffic-analysis methods. Moreover, two papers **do not fully negate** the importance of the field, and reviewer seems to have over-interpreted the original message.
>
> SIGCOMM’25 became publicly available online on **August 28** (the date list can be found at [SIGCOMM’25 conference site](https://conferences.sigcomm.org/sigcomm/2025/)), with the article listed as published on **August 27** according to the [ACM Digital Library](https://dl.acm.org/doi/abs/10.1145/3718958.3750498). Since the ICLR 2026 submission deadline was **September 24**, it could not be included in time. Additionally, according to the **ICLR Reviewer Guide** FAQ ([ICLR Reviewer Guide](https://iclr.cc/Conferences/2026/ReviewerGuide)), any paper published after **July 24** is considered contemporaneous.
>
> We will add and discuss these work in the revised version. Specifically, we will introduce CCS’22 and USENIX Sec’22 when discussing the pitfalls in ML, and SIGCOMM’25 when discussing pretraining methods, both of which will be covered in the **Related Work** section. Additionally, some of the methods we use align with those proposed in SIGCOMM’25 to mitigate these pitfalls, and we will highlight this and cite the paper accordingly. Finally, we have also added experiments to show how the performance of our model changes under **frozen** and **unfrozen fine-tuning**, which will be presented in the **Experiment** section.
>
> ****
>
> **For "Incremental and no methodological contribution"**, I do not agree with your assessment that “Incremental and no methodological contribution”, our contributions are not only in the engineering aspects but also lie in proposing a completely novel and systematic design methodology tailored to the unique characteristics of network traffic modeling.
>
> First, we did not simply transfer BERT. In response to the flaws in current traffic unit designs (such as fixed packet windows or treating burst traffic as a single packet), we introduced Flowlet, which dynamically segments packet sequences to better capture long-term structure and inter-packet relationships.
>
> Second, directly applying 4 Hex token has its problems. Multiple header fields are merged into a single token, losing the semantic boundaries between fields. To address this, we proposed Field Tokenization, segmenting based on protocol header fields and completely removing subword tokenization, ensuring a clearer representation of protocol semantics.
>
> Finally, our Protocol Stack Alignment Embedding Layer and two pretraining tasks are built on flowlet and field tokenization, enhancing the model’s ability to learn protocol layer semantics and inter-packet dependencies, thus addressing the challenge of multi-granular, multi-level semantic modeling in network traffic.
>
> ****
>
> **For "Flowlet Lightweight engineering contribution"**, our design of Flowlets is a new data representation approach. In network traffic modeling, traditional methods such as fixed-size windows or Burst-based segmentation have limitations, especially when dealing with long sequence data. Fixed-size windows often fail to effectively capture global structure, while Burst-based methods, in over 60% of cases, simplify the data into single packets (Figure 1a), thus losing cross-packet semantic information. Therefore, we propose Flowlets as a new modeling unit, dynamically segmenting packet sequences based on inter-arrival times (IAT). This design not only retains the local semantics of each packet but also effectively covers the overall traffic structure in long temporal sequences, thereby improving the ability to capture dynamic traffic behavior.
>
> [CCS’22] Jacobs A S, Beltiukov R, Willinger W, et al. Ai/ml for network security: The emperor has no clothes[C]//Proceedings of the 2022 ACM SIGSAC Conference on Computer and Communications Security. 2022: 1537-1551.
>
> [USENIX Sec’22] Arp D, Quiring E, Pendlebury F, et al. Dos and don'ts of machine learning in computer security[C]//31st USENIX Security Symposium (USENIX Security 22). 2022: 3971-3988.
>
> [SIGCOMM’25] Zhao Y, Dettori G, Boffa M, et al. The Sweet Danger of Sugar: Debunking Representation Learning for Encrypted Traffic Classification[C]//Proceedings of the ACM SIGCOMM 2025 Conference. 2025: 296-310.

---

> > ### Author Response · Authors · 2025-11-18
> >
> > **For "Tokenizer Lightweight engineering contribution"**, we apologize for the misrepresentation in paper. In our description, we referred to converting raw packets into hexadecimal sequences and then splitting them, using hexadecimal strings for illustration, but this does not accurately describe our method. Instead, in our implementation, we first parse each protocol header field and then convert the field value into a hex-unit token. Even if the field length is less than 4 bits (e.g., a 3-bit IP flag), we treat it as a separate token. This has been implemented in our anonymous code repository (see `datasetgeneration/field_token.py`). We showed the code related to the extraction of the IP header (starts from line 30) , as shown below, which can confirm that we are not splitting from hexadecimal strings.
> >
> > ```
> >     hex_fields = ''
> >     ip_layer = packet[IP]
> >     hex_fields += hex(ip_layer.version)[2:] + ' '
> >     hex_fields += hex(ip_layer.ihl)[2:] + ' '
> >     hex_fields += hex(ip_layer.tos)[2:].zfill(2) + ' '
> >     hex_fields += hex(ip_layer.len)[2:].zfill(4) + ' '
> >     hex_fields += hex(ip_layer.id)[2:].zfill(4) + ' '
> >     hex_fields += hex(ip_layer.flags.value)[2:] + ' '
> >     hex_fields += hex(ip_layer.frag)[2:].zfill(4) + ' '
> >     hex_fields += hex(ip_layer.ttl)[2:].zfill(2) + ' '
> >     hex_fields += hex(ip_layer.proto)[2:].zfill(2) + ' '
> >     hex_fields += hex(ip_layer.chksum)[2:].zfill(4) + ' '
> > ```
> >
> > We will correct the wording in the revision to match our implementation details accurately.
> >
> > Moreover, our method is **not "adding sub-16 encodings to a 65k vocabulary"**. We deliberately avoid further token splitting (e.g., BPE, WordPiece). In the case of traffic corpora, such NLP-style tokenizers only segment less than 1% of tokens, which in practice provides no benefit for traffic analysis and, more importantly, disrupts protocol-field semantics. Our approach keeps protocol fields as indivisible tokens.
> >
> > Finally, we beg to differ with the suggestion to isolate field tokenization in a standalone ablation experiment to verify its semantic contribution. In our view, flowlet segmentation and field tokenization together constitute our traffic representation, and therefore we did not originally separate them for ablation. However, in response to the reviewer’s request, we conducted an additional ablation study focusing solely on field tokenization in ISCXVPN(APP). The results show that removing field tokenization and using 4-hex and subword tokenization instead actually degrades model performance, confirming the necessity of our design.
> >
> > |                        |     AC     |     PR     |     RC     |     F1     |
> > | :--------------------: | :--------: | :--------: | :--------: | :--------: |
> > | w/o Field Tokenization |   0.7994   |   0.7670   |   0.7319   |   0.7396   |
> > |     Flowletformer      | **0.8328** | **0.7859** | **0.7507** | **0.7553** |
> >
> > **For "Possibly biased, weak evaluation, which translate into fundamentally flawed study"**, I am familiar with the CCS’22, USENIX’22, and SIGCOMM’25 findings, and I fully acknowledge the limitations they identify regarding learning directly from encrypted payloads.
> >
> > In our work, the modeling of encrypted payloads occurs primarily within the MFM pretraining task. But they are not the primary target of learning or modeling. In MFM, 50% of masked tokens are taken from semantically rich header fields; the remaining 50% are randomly selected, and about 25% of these also come from header fields. Only around 25% of masked tokens correspond to payload bytes. Therefore, the model does not fully rely on encrypted payload modeling, and the dominant learning signal comes from protocol-structure semantics.
> >
> > Regarding why we do not remove payload tokens entirely, there are three reasons:
> >
> > - Presence of unencrypted packets in the pretraining data. Real-world traces contain some unencrypted packets. Retaining payload helps the model generalize and leverage meaningful information when available.
> > - As part of the TCP payload, the TLS header contains a lot of useful information (such as certificate information and SNI, which are readable).
> > - Encrypted payloads are not perfectly random. As shown in [1], Table 5 and Figure 3 (Section 4.4), ideally encrypted data should exhibit maximum-entropy randomness, but actual implementations show varying levels of randomness. Their 15 statistical tests reveal that common encryption algorithms do not achieve perfect randomness, meaning ciphertext still contains detectable statistical patterns. Retaining payload allows the model to capture these residual characteristics without relying on semantic leakage.
> >
> > For these reasons, we retain payload tokens during pretraining, while ensuring that the vast majority of the learning signal comes from protocol fields rather than encrypted content. We will add and discuss these in the revised version.
> >
> > [1] Et-bert: A contextualized datagram representation with pre-training transformers for encrypted traffic classification.

---

> > > ### Author Response · Authors · 2025-11-18
> > >
> > > **For "Lack of statistical relevance"**, most prior works also report results with four decimal places and without confidence intervals or statistical tests, as in ET-BERT, TrafficFormer and YaTC. Even SIGCOMM’25 reports only one decimal place and likewise provides no statistical significance analysis. Moreover, in line 323 of our paper, we clearly state that our reported numbers are the **average of multiple runs**.
> > >
> > > **For "Lack of critical results analysis"**, we clarify that **Table 5 does not evaluate downstream classification tasks**. Instead, it measures the model’s field reconstruction ability, i.e., whether the model truly understands network traffic semantics.
> > >
> > > In this test, we deliberately mask key fields of a packet—such as source/destination IP and port, transport protocol, sequence/ack numbers, and TCP flags—and evaluate whether the model can recover the masked token using only the context of surrounding packets. This experiment examines whether the model has learned **real protocol-level patterns and field relationships**, not shortcut features.
> > >
> > > For example, when source/destination IP & port are masked, the model must infer them from the connection state observed in neighboring packets. This is genuine protocol reasoning, not shortcut learning of the type described in CCS’22.
> > >
> > > We will refine the description of this experiment in the revised version to avoid misunderstanding and to clearly distinguish it from downstream classification evaluation.
> > >
> > > **For "lack of comparison baselines benchmnark"**, we are not fully certain what the reviewer means by “ML for SEQNO sequence.” If this refers to standard ML models—including those capable of simple sequence modeling—then such baselines (e.g., DT, RF, XGBoost) are already included in Tables 1 and 2.
> > >
> > > If the comment refers to the implicit flow-ID shortcuts discussed in SIGCOMM’25 (e.g., using SEQNO or similar identifiers), this issue does not arise in our setting. In all experiments, **all packets from the same flow are placed entirely within the same train/val/test split**, ensuring that no implicit flow identifiers can leak across splits. Therefore, the shortcut mechanism identified in SIGCOMM’25 is explicitly eliminated in our evaluation.
> > >
> > > **For "lack of relevant technical details"**, I have reviewed the fine-tuning discussion in SIGCOMM’25. While I agree that their unfrozen experiments are informative, I do not fully share their conclusion. In our work, fine-tuning is performed in an **unfrozen (end-to-end)** manner.
> > >
> > > If end-to-end fine-tuning indeed “destroys” pretrained knowledge, as implied by the reviewer, then a model trained from **random initialization** should exhibit comparable performance to the pretrained model with all layers unfrozen. Yet, our ablation (Table 3), along with ablation in prior work like ET-BERT , shows that removing pretraining leads to a substantial degradation in downstream task performance. This demonstrates that the pretrained knowledge is not overwritten by end-to-end fine-tuning but instead provides essential initialization for downstream adaptation.
> > >
> > > Additionally, we conducted further experiments using unfrozen fine-tuning, and the results are shown below. Both approaches have their own merits.  For most datasets, the performance under the **frozen** setting does not show a noticeable decline, indicating that the knowledge acquired during pre-training can be effectively utilized by downstream tasks. The only exception is ISCX-Tor2016. Since the pre-training corpus contains almost no Tor traffic, the model lacks the necessary prior knowledge, resulting in a significant performance drop when parameters are frozen. This suggests that pre-training data should be more diverse, and that **unfrozen fine-tuning is necessary** when encountering traffic types that are absent from the pre-training corpus.
> > >
> > > | ISCX-VPN(Service) |   AC   |   F1   |     Browser     |   AC   |   F1   |
> > > | :---------------: | :----: | :----: | :-------------: | :----: | :----: |
> > > |      Frozen       | 0.8645 | 0.8446 |     Frozen      | 0.6583 | 0.6616 |
> > > |     Unfrozen      | 0.9578 | 0.9493 |    Unfrozen     | 0.7083 | 0.6932 |
> > > | **ISCX-Tor2016**  | **AC** | **F1** |  **USTC-TFC**   | **AC** | **F1** |
> > > |      Frozen       | 0.5632 | 0.3931 |     Frozen      | 0.9563 | 0.9568 |
> > > |     Unfrozen      | 0.9078 | 0.8436 |    Unfrozen     | 0.9742 | 0.9741 |
> > > | **ISCX-VPN(APP)** | **AC** | **F1** | **CIC-IDS2017** | **AC** | **F1** |
> > > |      Frozen       | 0.7893 | 0.7181 |     Frozen      | 0.8778 | 0.8683 |
> > > |     Unfrozen      | 0.8328 | 0.7553 |    Unfrozen     | 0.9200 | 0.9109 |
> > > |  **CSTNET-TLS**   | **AC** | **F1** | **CIC-IoT2022** | **AC** | **F1** |
> > > |      Frozen       | 0.6151 | 0.5614 |     Frozen      | 0.7969 | 0.7402 |
> > > |     Unfrozen      | 0.8518 | 0.8377 |    Unfrozen     | 0,9177 | 0.8808 |

---

> ### Comment · Reviewer_LKds · 2025-11-27
> **noted author rebuttal**
>
> I have noted the reviewers answer.  Particularly,
>
> 0) the absence of statistical relevance  -- saying other related literature does not conduct statistical relevant analysis is not a justification for not doing it ? also none of the cited work appear in ICLR, meaning that when submitting to ICLR authors should prepare a submission adhering to the target venue expectation
>
> """
> For "Lack of statistical relevance", most prior works also report results with four decimal places and without confidence intervals or statistical tests, as in ET-BERT, TrafficFormer and YaTC. Even SIGCOMM’25 reports only one decimal place and likewise provides no statistical significance analysis. Moreover, in line 323 of our paper, we clearly state that our reported numbers are the average of multiple runs.
> """
>
> 1) I value the qualitative clarification on the  tokenization  (as per above, the quantitative assessment is still plagued by lack of statistical relevance).
>
>                                               AC        	PR	        RC      	F1
>        w/o Field Tokenization	0.7994	0.7670	0.7319	0.7396
>        Flowletformer          	0.8328	0.7859	0.7507	0.7553
>
>
> 2)  while I agree that weight initialization starting from  "Foundational pre-traning" is different than random, authors agree that end-to-end fine-tuning indeed “destroys” pretrained knowledge.  To me, this is the biggest weakness of this paper, especially because ICLR stands for International Conference on Learning Representations -- which authors admit to need overwriting  because essentially the architecture is not able to learn via pre-training a representation with sufficient discriminative powerful, that a simple model head could exploit. This may be OK for venues such as where the original papers ET-BERT, YaTC etc were published, but is surely not good enough for IC"LR" as the method seems to focus on the downstream task, more than any actual learning process/method
>
> """ If end-to-end fine-tuning indeed “destroys” pretrained knowledge, as implied by the reviewer, then a model trained from random initialization should exhibit comparable performance to the pretrained model with all layers unfrozen. Yet, our ablation (Table 3), along with ablation in prior work like ET-BERT , shows that removing pretraining leads to a substantial degradation in downstream task performance. This demonstrates that the pretrained knowledge is not overwritten by end-to-end fine-tuning but instead provides essential initialization for downstream adaptation.
>
> 3) as a consequence of 3,  you end up having N models one fit to each dataset, instead of having 1 True Foundation Backbone + one head for each dataset; so even in terms of practical downstream deployment, the landing point is not the most desirable one
>
> as such, and mostly for points 0) and 2), I maintain my judgement

---

> > ### Author Response · Authors · 2025-12-02
> > **Our Response to the Reviewer LKds (1/2)**
> >
> > We thank the reviewer for the response and the additional questions. We address them point by point:
> > ****
> >
> > **For Points 0 and 1, Thank you for highlighting the need for statistical validation.**
> >
> > Building on the original three runs, we conducted **7 additional runs** for all pretraining methods across all **8 datasets**, using seeds **101–107**. This gives us **10 runs per dataset**, on which we performed **paired t-tests** using the F1 scores.
> >
> > The table reports the **mean F1 score and its standard deviation** for each method. The results are summarized below.
> >
> > |        Dataset        |     ETBERT      |  TrafficFormer  |      YaTC       |    FLowletFormer    |
> > | :-------------------: | :-------------: | :-------------: | :-------------: | :-----------------: |
> > | **ISCX-VPN(Service)** | 0.7853 ± 0.0701 | 0.8412 ± 0.0323 | 0.8953 ± 0.0132 | **0.9358 ± 0.0325** |
> > |                       |     p<0.001     |     p<0.001     |     p=0.005     |          -          |
> > |   **ISCX-Tor2016**    | 0.6799 ± 0.0398 | 0.7076 ± 0.0413 | 0.7205 ± 0.0384 | **0.8121 ± 0.0497** |
> > |                       |     p<0.001     |     p<0.001     |     p<0.001     |          -          |
> > |   **ISCX-VPN(APP)**   | 0.6958 ± 0.0163 | 0.7229 ± 0.0212 | 0.7360 ± 0.0172 | **0.7629 ± 0.0191** |
> > |                       |     p<0.001     |     p=0.002     |     p=0.015     |          -          |
> > |    **CSTNET-TLS**     | 0.7493 ± 0.0237 | 0.7514 ± 0.0125 | 0.8215 ± 0.0051 | **0.8347 ± 0.0079** |
> > |                       |     p<0.001     |     p<0.001     |     p=0.004     |          -          |
> > |      **Browser**      | 0.2993 ± 0.1113 | 0.3717 ± 0.1328 | 0.5083 ± 0.0529 | **0.6849 ± 0.0406** |
> > |                       |     p<0.001     |     p<0.001     |     p<0.001     |          -          |
> > |     **USTC-TFC**      | 0.9711 ± 0.0036 | 0.9699 ± 0.0061 | 0.9710 ± 0.0052 |   0.9753 ± 0.0053   |
> > |                       |     p=0.045     |     p=0.039     |     p=0.051     |          -          |
> > |    **CIC-IDS2017**    | 0.8496 ± 0.0244 | 0.8619 ± 0.0197 | 0.8959 ± 0.0148 |   0.9038 ± 0.0083   |
> > |                       |     p<0.001     |     p<0.001     |     p=0.115     |          -          |
> > |    **CIC-IoT202**     | 0.8103 ± 0.0145 | 0.8275 ± 0.0075 | 0.8269 ± 0.0253 | **0.8751 ± 0.0191** |
> > |                       |     p<0.001     |     p<0.001     |     p<0.001     |          -          |
> >
> > Overall, the analysis shows that **on 6 out of 8 datasets, our method is significantly better** than all three baselines.
> >
> > On **USTC-TFC** and **CIC-IDS2017**, our method is **significantly better than ET-BERT and TrafficFormer**, while showing **no significant difference compared with YaTC**. However, **our average F1 score on both datasets remains higher** than YaTC.
> >
> > Although time constraints prevented us from running significance tests for all experiments, the current results already provide strong preliminary evidence of our method’s effectiveness, and we will include full significance analysis in the appendix of the revised version.
> >
> > ****
> >
> > **For Points 2 and 3, we thank the reviewer for the thoughtful comments regarding downstream fine-tuning strategies.**
> >
> > First, we would like to clarify that our previous mention of “end-to-end fine-tuning destroys pretrained knowledge” was only a summary of the reviewer’s viewpoint, not an endorsement of that claim.
> >
> > In fact, extensive NLP literature provides evidence that fine-tuning does **not** “destroy” pretrained representations:
> >
> > **(1). ACL 2022 [1] explicitly shows that:**
> >
> > > ​	*Fine-tuning does not introduce arbitrary changes to representations; instead, it adjusts the representations to downstream tasks while largely preserving the original spatial structure of the data points.*
> >
> > Using geometric structure probes (DIRECT PROBE) and frozen-representation classifiers across multiple NLP tasks (POS tagging, dependency parsing, preposition sense disambiguation, text classification), the authors demonstrate that fine-tuning:
> >
> > - does **not** reconstruct or overwrite the representation space;
> > - instead makes previously ambiguous class regions **more separable and more clustered**;
> >
> > - thereby improving downstream task performance.
> >
> > **(2). Other study [2] shows that:** For most tasks, **frozen and unfrozen fine-tuning yield similar performance**.
> >
> > There is no universal rule for which method is always superior. The choice should depend on dataset size and task characteristics. Therefore, we do not agree with the absolute claim in SIGCOMM 2025 [3] that unfrozen fine-tuning is fundamentally undesirable.
> >
> > [1] Zhou Y, Srikumar V. A closer look at how fine-tuning changes BERT[C]//Proceedings of the ACL.
> >
> > [2] Matthew E. Peters, et.al. To Tune or Not to Tune? Adapting Pretrained Representations to Diverse Tasks. Proceedings of the Workshop on RepL4NLP-2019. ACL.
> >
> > [3] Zhao Y, et al. The Sweet Danger of Sugar: Debunking Representation Learning for Encrypted Traffic Classification[C] // Proceedings of the ACM SIGCOMM.

---

> > > ### Author Response · Authors · 2025-12-02
> > > **Our Response to the Reviewer LKds (2/2)**
> > >
> > > Second, as shown in the table of `Official Comment by Authors #4`, FlowletFormer does learn discriminative and structurally meaningful representations during pretraining, that even a **simple MLP classification header** can effectively utilize them. We include the table again here for convenience. From the table, we observe that on five datasets (ISCX-VPN (App), Browser, USTC-TFC and CIC-IDS2017), the F1 score of the frozen model drops by only about 4% compared to the unfrozen model. This level of degradation is fully consistent with observations reported in SIGCOMM 2025 [3] (e.g., Table 4 of [3], where VPNApp drops from 74.8% to 71.0% and TLS-120 drops from 69.2% to 63.7% in F1 when switching to frozen fine-tuning).
> > >
> > > For the remaining three datasets, the trends can be explained by the characteristics of the pretraining corpus.
> > >
> > > - On ISCX-VPN (Service), although the frozen model shows a ~10% drop, its F1 score remains at a strong level, indicating that the representation is still robust.
> > > - The performance drops observed on ISCX-Tor2016, CSTNET-TLS, and CIC-IoT2022 are expected, as the pretraining corpus contains **no Tor traffic** and only **a limited amount of TLS 1.3 and IoT** packets. Consequently, the absence of relevant pretraining signals naturally constrains the quality of the frozen representations on these datasets.
> > >
> > > Overall, the observed performance differences are reasonable and align with prior findings, and most datasets show only minor degradation under the frozen setting.
> > >
> > > ****
> > >
> > > To further address the reviewer’s concerns, we conducted an additional experiments:
> > >
> > > - **Stronger classification head: two-layer MLP (aligned with SIGCOMM 2025 [3])**.
> > >
> > > In our earlier frozen experiments, we used a **one-layer MLP** classifier, but SIGCOMM 2025 [3] used a **two-layer MLP**. We also evaluate the frozen setting using the same two-layer MLP classifier with SIGCOMM 2025 in FlowletFormer. (To clarify, the “layer” here refer to **hidden layers** in the MLP classifier.)
> > >
> > > |   ISCX-VPN(Service)   |   AC   |   F1   |        Browser        |   AC   |   F1   |
> > > | :-------------------: | :----: | :----: | :-------------------: | :----: | :----: |
> > > | 2-layer-header+Frozen | 0.8733 | 0.8534 | 2-layer-header+Frozen | 0.6750 | 0.7084 |
> > > |        Frozen         | 0.8645 | 0.8446 |        Frozen         | 0.6583 | 0.6616 |
> > > |       Unfrozen        | 0.9578 | 0.9493 |       Unfrozen        | 0.7083 | 0.6932 |
> > > |   **ISCX-Tor2016**    | **AC** | **F1** |     **USTC-TFC**      | **AC** | **F1** |
> > > | 2-layer-header+Frozen | 0.5939 | 0.4301 | 2-layer-header+Frozen | 0.9700 | 0.9703 |
> > > |        Frozen         | 0.5632 | 0.3931 |        Frozen         | 0.9563 | 0.9568 |
> > > |       Unfrozen        | 0.9078 | 0.8436 |       Unfrozen        | 0.9742 | 0.9741 |
> > > |   **ISCX-VPN(APP)**   | **AC** | **F1** |    **CIC-IDS2017**    | **AC** | **F1** |
> > > | 2-layer-header+Frozen | 0.8055 | 0.7372 | 2-layer-header+Frozen | 0.9017 | 0.8899 |
> > > |        Frozen         | 0.7893 | 0.7181 |        Frozen         | 0.8778 | 0.8683 |
> > > |       Unfrozen        | 0.8328 | 0.7553 |       Unfrozen        | 0.9200 | 0.9109 |
> > > |    **CSTNET-TLS**     | **AC** | **F1** |    **CIC-IoT2022**    | **AC** | **F1** |
> > > | 2-layer-header+Frozen | 0.6843 | 0.6494 | 2-layer-header+Frozen | 0.8684 | 0.8320 |
> > > |        Frozen         | 0.6151 | 0.5614 |        Frozen         | 0.7969 | 0.7402 |
> > > |       Unfrozen        | 0.8518 | 0.8377 |       Unfrozen        | 0,9177 | 0.8808 |
> > >
> > > We find that replacing the simple MLP head with a more expressive two-layer MLP indeed improves the frozen performance.
> > > For the datasets on which FlowletFormer already performed well under the frozen setting (ISCX-VPN (Service), ISCX-VPN (App), Browser, USTC-TFC, CIC-IDS2017), the F1 score increases by **1%–4%**.
> > >
> > > For datasets that contain traffic types **underrepresented or nearly absent** in the pretraining dataset (ISCX-Tor2016, CSTNET-TLS, and CIC-IoT2022), the improvements are more pronounced:
> > >
> > > - **ISCX-Tor2016:** +5%
> > > - **CSTNET-TLS:** +8%
> > > - **CIC-IoT2022:** +9%
> > >
> > > A straightforward comparison is that both our method and the SIGCOMM 2025 [3] evaluate on the CSTNET dataset, and our result is on par with theirs (64.9 (ours) vs. 63.7).
> > >
> > > These results further confirm that FlowletFormer’s pretrained representations are structurally rich, and stronger classifier heads are able to exploit this representations more effectively.
> > >
> > > ****
> > >
> > > We again thank the reviewer for the constructive feedback. We hope that the additional statistical analyses, stronger evidence on representation quality, and newly added experiments help address the concerns raised. We will incorporate all clarified points and results into the revised version.
> > >
> > > [3] Zhao Y, et al. The Sweet Danger of Sugar: Debunking Representation Learning for Encrypted Traffic Classification[C] // Proceedings of the ACM SIGCOMM.

---

### Official Review · Reviewer_FjKy · 2025-10-31

**Soundness:** 2
**Presentation:** 3
**Contribution:** 2
**Rating:** 6
**Confidence:** 3

**Summary:**

The paper introduces FlowletFormer, a Flowlet-based pre-training framework for network traffic analysis that captures cross-packet context, protocol semantics, and flow-level behavior. It employs Flowlet and Field Tokenization, a Protocol Stack Alignment Embedding Layer, and two Flowlet-inspired pre-training tasks to enhance semantic and behavioral understanding. Experiments demonstrate that FlowletFormer achieves superior accuracy, few-shot adaptability, and robustness by incorporating domain-specific network knowledge.

**Strengths:**

- The paper introduces Flowlet and  Field Tokenization as efficient traffic representation.
- Protocol Stack Alignment-Based Embedding Layer is proposed to explicitly encode the hierarchical semantics of network protocols, enabling the model to distinguish fields across protocol boundaries and better capture protocol-specific behaviors.
- Two pre-training tasks are proposed. Extensive experiments are performed on comprehensive downstream tasks under various settings, demonstrating the effectiveness of the proposed method.
- The paper is overall well written.

**Weaknesses:**

- Besides the superior performance, the technical novelty follows the general BERT pretraining pipeline with similar pretraining tasks.
- From the development of the community, besides the pretraining code, the model weights are suggested to be open-sourced.

**Questions:**

- In Table 2, Flowlet Former underperforms TrafficFormer on USTC-TFC only. Please explain the reason.

---

> ### Author Response · Authors · 2025-11-18
>
> We sincerely thank the reviewer for the constructive comments and helpful questions. We respond to each point in detail below.
>
> ****
>
> **For Weakness 1**, we understand that the reviewer perceives our work similar to standard BERT pretraining. We would like to clarify that the novelty of FlowletFormer does not lie in modifying the BERT backbone, but in introducing a traffic-native representation and pretraining framework that cannot be obtained by directly applying NLP-style BERT to traffic.
>
> 1. Traffic-specific representation, not a direct BERT adaptation.
>    FlowletFormer is built upon a **representation model** designed specifically for network traffic, including Flowlet and Field Tokenization. These components define the semantic units of traffic by preserving protocol boundaries and temporal correlations—properties that existing 4-hex/subword or packet-window methods cannot capture.
> 2. The three components are **interdependent and jointly** enable the pretraining tasks.
>    Our Protocol Stack Alignment (PSA) Embedding and both pretraining tasks (Masked Field Model and Flowlet Prediction Task) rely directly on this representation. Without field tokenization, MFM cannot target semantic header fields; without flowlets, FPT cannot model behaviorally coherent interactions.
> 3. Contribution is in the representation + embedding + task triad.
>    The key methodological innovation lies in rethinking how traffic should be represented, embedded, and pre-trained. This triad is tailored to the hierarchical and behavioral nature of network traffic and differs from repurposing generic BERT pipelines.
>
> ****
>
> **For Weakness 2**, We fully agree with the reviewer on the importance of open-source availability. We will release **the pretrained FlowletFormer model weights** besides the pretraining and finetuning code to support reproducibility and benefit the research community.
>
> ****
>
> **For Question 1**, thank you for the careful observation regarding the 0.1% lower F1 score compared to TrafficFormer on the Malware dataset. We agree this specific result. But, we think it is best understood in the context of FlowletFormer's overall exceptional performance.
>
> Overall, FlowletFormer outperformed all baselines on **7 out of the 8** diverse datasets we evaluated (Tables 1 and 2), and in many other tests its performance was also superior to TrafficFormer. This demonstrates the general effectiveness and broad applicability of our approach across various traffic types (e.g., VPN, IoT, malicious traffic, general traffic). several of the other datasets we used, such as **CICIDS2017**, also contain **malware or attack traffic**, where FlowletFormer likewise achieved competitive or superior accuracy. This further indicates that our method does **not** exhibit inherent weaknesses in identifying malicious or abnormal traffic.
>
> Regarding the observed small gap (e.g., 0.1% difference in F1 score), we believe this can be attributed to many factors. As detailed in the paper, we conducted multiple rounds of experiments with different random seeds. The table below shows the F1 scores from three rounds of experiments, where in two cases, FlowletFormer either outperforms or matches TrafficFormer on the Malware dataset:
>
> |  Seed  | TrafficFormer | FlowletFormer |
> | :----: | :-----------: | :-----------: |
> | SEED 1 |  **0.9746**   |    0.9648     |
> | SEED 2 |    0.9751     |  **0.9776**   |
> | SEED 3 |    0.9778     |  **0.9800**   |
>
> In addition to random seeds, **dataset splitting** can also introduce slight variations. To verify this, we conducted experiments with **5 different random splits**, changing the random_seed each time. The table below shows the results, indicating that in **3 out of 5 splits**, FlowletFormer either outperforms or matches TrafficFormer on the Malware dataset:
>
> |        | TrafficFormer | FlowletFormer |
> | :----: | :-----------: | :-----------: |
> | SEED 1 |    0.9753     |  **0.9962**   |
> | SEED 2 |  **0.9900**   |    0.9615     |
> | SEED 3 |    0.9764     |  **0.9789**   |
> | SEED 4 |    0.9740     |  **0.9776**   |
> | SEED 5 |  **0.9788**   |    0.9776     |
>
> ****
>
> Thank you once again for your thorough and constructive feedback and we hope our responses have been helpful in clarifying our work.

---

### Official Review · Reviewer_WPqN · 2025-11-09

**Soundness:** 2
**Presentation:** 3
**Contribution:** 2
**Rating:** 6
**Confidence:** 2

**Summary:**

This paper addresses the problem of network traffic analysis. To effectively capture cross-packet context, protocol-aware structures, and flow-level behaviors, the authors propose a Flowlet-based pre-training framework. The framework consists of three major components: Flowlet and Field Tokenization, a Protocol Stack Alignment Embedding Layer, and two pre-training tasks designed to enhance both intra-packet understanding and inter-flow learning.
Experimental results demonstrate the effectiveness and robustness of the proposed approach.

**Strengths:**

The overall paper is well-written.

This work investigates a BERT-based pre-training model, called FlowletFormer, for network traffic analysis. The proposed framework incorporates three key strategies: (1) Flowlet segmentation, (2) a Protocol Stack Alignment-based Embedding Layer, and (3) two pre-training tasks—Masked Field Modeling (MFM) and Flowlet Prediction Task (FPT).
The effectiveness of the method is evaluated on eight public datasets, demonstrating the superiority and robustness of the proposed approach.

**Weaknesses:**

The three components together constitute FlowletFormer. Among them, which element plays the most critical role, and how are these components interrelated?

In the FlowletFormer framework, how are the Flowlet Prediction Task (FPT) and the Masked Field Modeling (MFM) task distinguished in the flowchart?

Regarding the downstream tasks, the seven tasks mentioned appear to be from previous benchmarks. Have any new tasks been introduced or conducted in this work?

It seems somewhat unusual that the performance of TrafficFormer (2025) is lower than that of YaTC (2023) and ET-BERT (2022) in most cases—what factors might explain this discrepancy?

Finally, in the ablation study, the pre-training (PT) stage shows the greatest influence on performance across the four evaluation metrics compared to other components. Could you clarify how the pre-training data were sourced from the three repositories?

**Questions:**

see above weakness

---

> ### Author Response · Authors · 2025-11-18
>
> We sincerely thank the reviewer for the constructive and insightful feedback. We address each comment in detail below.
>
> ****
>
> **For Weakness 1**, you are absolutely correct that FlowletFormer is composed of three core components. Among them, we consider Flowlet and Field Tokenization to be the most critical. The reason is that both Protocol Stack Alignment (PSA) Embedding Layer and the two pre-training tasks rely on fine-grained field-level tokens. If the commonly used 4-Hex tokenization is adopted, multiple header fields will be merged into a single token, making PSA embedding and Masked Field Modeling (MFM) unable to function as intended.
>
> These components work in a progressive, mutually reinforcing manner. Flowlet segmentation groups packets belonging to the same logical interaction (e.g., HTTP request–response), exposing flow-level temporal dependencies. The PSA embedding layer preserves and aligns protocol-layer semantics in token space, offering structured priors. The two pre-training tasks then enhance representation learning from different perspectives: MFM strengthens intra-packet field reasoning, while the Flowlet Prediction Task (FPT) models temporal and behavioral patterns across flowlets.
>
> Our ablation study (Table 3) shows that removing any of the three components results in a performance drop.
>
> ****
>
> **For Weakness 2**, FPT predicts the relationship between adjacent flowlets using H[CLS], while MFM predicts the masked fields corresponding to H[MASK] in the figure. Accordingly, in our workflow diagram (Figure 3), the arrow from H[CLS] points to the FPT branch, and the arrow from H[MASK] points to the MFM branch, clearly separating the two operation paths.
>
> We appreciate the reviewer’s valuable suggestion. In the revised version, we will apply clearer color coding and add explicit annotations to make the distinction between MFM and FPT in Figure 2 more intuitive.
>
> ****
>
> **For Weakness 3**, the seven downstream tasks are consistent with prior work to ensure fair comparison. In addition to the seven downstream tasks, we introduced four **Field Understanding Tasks** (Table 5) to evaluate the model's ability to comprehend general traffic patterns. Specifically, we used **Word Analogies Similarity** (Table 6) to assess the model's understanding of field semantics. Furthermore, we tested the model's robustness to **corrupted traffic data** (Figure 4c).
>
> ****
>
> **For Weakness 4**, We acknowledge the reviewer’s observation regarding the relative performance of TrafficFormer. To ensure fairness, all baseline models—including ET-BERT, YaTC, and TrafficFormer—were pre-trained on exactly the same pre-training corpus and fine-tuned on the same downstream datasets, with identical training/validation/test splits in our experiments. This controlled setup may differ from the original TrafficFormer configuration and can therefore influence absolute performance numbers.
>
> In addition, the original TrafficFormer paper reports that its performance is comparable to ET-BERT and YaTC when evaluated without the proposed Traffic Data Augmentation (randomized reinitialization of header fields). In their Tables 5–7, TrafficFormer without augmentation indeed performs similarly to or slightly below these earlier models. The performance gains highlighted in the original paper come after applying this augmentation strategy.
>
> Since our comparison does not apply TrafficFormer’s data augmentation (to maintain consistent and fair evaluation across baselines), the results we observe are aligned with the findings reported in the original TrafficFormer paper.
>
> ****
>
> **For Weakness 5**, we thank the reviewer for this question. The model primarily learns traffic patterns and structural relationships during the pre-training stage; therefore, removing pre-training leads to the most significant performance degradation. Below, we provide a detailed explanation of how the pre-training data was collected and processed.
>
> The pre-training data was sourced from three repositories: ISCX-VPN2016 (NonVPN), CIC-IDS2017 (Monday), and the WIDE backbone dataset (January 1, 2024). It is important to emphasize that the downstream tasks use ISCX-VPN2016 (VPN) and CIC-IDS2017 (other days), ensuring that **there is no data reuse or leakage between pre-training and downstream task.**
>
> The processing procedure is as follows:
>
> First, all raw traffic files were converted into a unified PCAP format for consistent handling. We then extracted flows based on the five-tuple (source IP, destination IP, source port, destination port, protocol) and the rules defined in Table 8. Each flow was further segmented into flowlets using the inter-arrival-time (IAT) threshold. Finally, we paired flowlets according to the FPT construction method and applied token masking following the MFM strategy, resulting in the complete pre-training corpus.
>
> ****
>
> We hope these explanations provide the necessary clarity on our proposed methods. Thank you again for your valuable feedback.

---

### Author Response · Authors · 2025-11-26
**General Response of Submission 809**

We sincerely thank all reviewers for the thoughtful, detailed, and constructive feedback. To improve our work, we added the following improvements to our revised paper (highlighted in **blue**):

1. We redrew **Figure 3** in Section 3.2, highlighting the pre-training task branches with thicker and color-coded arrows to clearly distinguish MFM and FPT. More details are provided in *Official Comment by Authors to Reviewer WPqN* under `For Weakness 2`.
2. We clarified our plan to open-source the pretrained model weights. Due to storage limitations of the anonymous repository, the download link will be provided after the review period. Further details are included in *Official Comment by Authors to Reviewer FjKy* under `For Weakness 2`.
3. We added discussions of *CCS'22* and *USENIX Sec’22* in Section 2.1, and incorporated *SIGCOMM’25* into Sections 2.2, 3.4, and 4.7.
4. We revised the description of **Field Tokenization** in Section 3.1. The clarification and relevant code excerpts can be found in *Official Comment by Authors to Reviewer LKds 2* under `For "Tokenizer Lightweight engineering contribution"`.
5. We added Sections 3.4 and 4.7 to provide a detailed explanation of the **fine-tuning method** and to include new experiments comparing the *frozen* and *unfrozen* settings. Further discussion is provided in *Official Comment by Authors to Reviewer LKds 3* under `For "lack of relevant technical details"`.
6. We added the **w/o Field Tokenization** setting to the ablation study in Table 3. The detailed analysis is included in *Official Comment by Authors to Reviewer LKds 2* under `For "Tokenizer Lightweight engineering contribution"`.

We sincerely thank the reviewers for their valuable comments, which helped us significantly improve the clarity, completeness, and technical rigor of the paper. We look forward to further discussion and appreciate your further feedback.



[CCS’22] Jacobs A S, Beltiukov R, Willinger W, et al. Ai/ml for network security: The emperor has no clothes[C]//Proceedings of the 2022 ACM SIGSAC Conference on Computer and Communications Security. 2022: 1537-1551.

[USENIX Sec’22] Arp D, Quiring E, Pendlebury F, et al. Dos and don'ts of machine learning in computer security[C]//31st USENIX Security Symposium (USENIX Security 22). 2022: 3971-3988.

[SIGCOMM’25] Zhao Y, Dettori G, Boffa M, et al. The Sweet Danger of Sugar: Debunking Representation Learning for Encrypted Traffic Classification[C]//Proceedings of the ACM SIGCOMM 2025 Conference. 2025: 296-310.

---

### Meta-Review · Area_Chair_hmMW · 2026-01-06

**Summary:**

This paper proposes a pretrained model based traffic representation learning framework named FlowletFormer to capture the semantic and structural features of network traffic. The main concerns lie in the proposed framework showing minor novely compared to existing per-training methods. According to author responses and the revised manuscript, the concerns about novelty remain outstanding. Additionally, although the manuscript is titled “Rethinking”, the discussion of the limitations of existing methods is limited and lacks deeper insights. Therefore, my overall recommendation is Reject.

**Reviewer Concerns:**

The concerns about the technical details (reviewer FjKy, LKds) and about the experiments(reviewer WPqN, FjKy, LKds) are well answered in the rebuttal and the revised manuscript. However, the main concerns about novelty (reviewer FjKy, LKds) remain outstanding.

**Reviewer Scores:**

Reviewer WPqN: The concerns raised by the reviewer are detailedly answered. Hence, I think the reviewer WPqN might change his score.
Reviewer FjKy: The concern about novely of the proposed framework is hardly solved in the response of the authors. Hence, I think the reviewer FjKy probably will not change his score.
Reviewer LKds: Based on the current discussions, I think the reviewer LKds probably will not change his score.

---

### Decision · Program_Chairs · 2026-01-26

Reject